# PALQO: Physics-informed Model for Accelerating Large-scale Quantum Optimization

**Yiming Huang**[1*], **Yajie Hao**[2*], **Jing Zhou**[5], **Xiao Yuan**[1†], **Xiaoting Wang**[2†], **Yuxuan Du**[3,4†]

[1]Center on Frontiers of Computing Studies,
and School of Computer Science, Peking University, Beijing, China
[2]Institute of Fundamental and Frontier Sciences
University of Electronic Science and Technology of China, Chengdu, China
[3]College of Computing and Data Science
Nanyang Technological University, Singapore, Singapore
[4]School of Physical and Mathematical Sciences
Nanyang Technological University, Singapore, Singapore
[5]Department of Physics, Fudan University, Shanghai, China
yiminghwang@gmail.com, yuxuan.du@ntu.edu.sg, xiaoyuan@pku.edu.cn
xiaoting@uestc.edu.cn

## Abstract

Variational quantum algorithms (VQAs) are leading strategies to reach practical utilities of near-term quantum devices. However, the no-cloning theorem in quantum mechanics precludes standard backpropagation, leading to prohibitive quantum resource costs when applying VQAs to large-scale tasks. To address this challenge, we reformulate the training dynamics of VQAs as a nonlinear partial differential equation and propose a novel protocol that leverages physics-informed neural networks (PINNs) to model this dynamical system efficiently. Given a small amount of training trajectory data collected from quantum devices, our protocol predicts the parameter updates of VQAs over multiple iterations on the classical side, dramatically reducing quantum resource costs. Through systematic numerical experiments, we demonstrate that our method achieves up to a 30x speedup compared to conventional methods and reduces quantum resource costs by as much as 90% for tasks involving up to 40 qubits, including ground state preparation of different quantum systems, while maintaining competitive accuracy. Our approach complements existing techniques aimed at improving the efficiency of VQAs and further strengthens their potential for practical applications.

## 1 Introduction

Modern quantum computers, with a steadily increasing number of high-quality qubits, are approaching the threshold of practical utility [1–3]. In this pursuit, variational quantum algorithms (VQAs) [4–12] have emerged as a leading strategy, attributed to their flexibility in accommodating circuit depth and qubit connectivity among different platforms. In recent years, a wide range of theoretical and experimental studies have demonstrated the feasibility of VQAs across diverse applications, such as quantum chemistry [13–15], optimization [16–18], and machine learning [19–23]. Despite the progress, they face critical challenges when applied to large-scale problems. In particular, the no-cloning theorem and the unitary constraints in the quantum universe prohibit the use of backpropagation techniques common in deep learning [24], requiring VQAs to update sequentially to

---

*Equal contribution.
†Corresponding authors.

39th Conference on Neural Information Processing Systems (NeurIPS 2025).

minimize a predefined cost function [25, 26]. This approach imposes substantial and even prohibitive quantum resource demands, especially in terms of the number of measurements required. Given the scarcity of quantum computers in the foreseeable future, enhancing the optimization efficiency of VQAs while minimizing resource consumption is crucial for enabling their practical deployment.

To advance VQAs for large-scale problems, substantial efforts have been devoted to improving the optimization efficiency. Prior literature in this field can be broadly categorized into three primary classes (see Sec. 2.1 for details). The first aims to reduce measurement costs in quantum many-body and chemistry problems by grouping terms in the Hamiltonian to enable simultaneous measurements [27–29]. The second harnesses classical simulators or learning models to identify well-initialized parameters that are close to local minima of the cost landscape for a given VQA, thereby improving convergence efficiency [30–32]. The third seeks to predict the dynamics of parameter updates by revising past optimization trajectories for the given task [33], as exemplified by methods such as recurrent neural networks [34, 35] and QuACK [36]. Despite significant advancements, no existing approach effectively balances optimization efficiency and accuracy at scale. In this regard, a critical question arises: *is it possible to achieve both for large-scale VQA systems?*

Towards this question, we observe that prior efforts have primarily focused on hardware improvements and heuristic optimizations, while the potential of approximating training dynamics to alleviate the quantum resource burden **remains underexplored**. In response, we introduce a fresh perspective by utilizing Taylor expansion to reformulate the parameter optimization process in VQAs as a nonlinear partial differential equation (PDE). In this way, the evolution of this nonlinear PDE corresponds to the trajectory of parameter updates during training. Building on this formulation, we devise a protocol, dubbed PALQO, that employs a physics-informed neural network (PINN) [37, 38] to approximate solutions to the PDE, where high-order terms in the Taylor expansion serve as boundary conditions. The proposed framework is generic, which encompasses the quantum neural tangent kernel (QNTK) as a special case [39–41]. Besides, we prove that a polynomial number of training samples is sufficient to ensure that PALQO attains a satisfactory generalization ability.

We then conduct extensive numerical simulations on different ground state preparation tasks, including the transverse-field Ising model, Heisenberg model, and multiple molecule systems, to investigate the effectiveness of PALQO. Simulation results up to 40 qubits indicate that by learning from a limited set of initial parameter updates obtained from quantum devices, PALQO, which is deployed on classical hardware, can accurately predict future parameter updates. These results indicate the effectiveness of reducing the number of quantum measurements required during optimization. Numerical experiments on ground state preparation tasks across large-scale systems, involving up to 40 qubits, validate the effectiveness of PALQO. Moreover, we show that the proposed PALQO is complementary to existing approaches for improving the optimization efficiency of VQAs. To support the community, we release our source code at [42]. These results open a new avenue for leveraging the power of PINNs to enhance the efficiency of VQAs and advance the frontier of practical quantum computing.

**Contributions**. For clarity, we summarize our main contributions below. **(1)** To our best knowledge, we establish the first general framework between the optimization trajectory of VQAs and PDE, thereby enabling the employment of various PINNs to advance the optimization efficiency of large-scale VQAs. **(2)** We propose PALQO, an effective PINN oriented to reduce the required number of measurements when training large-scale VQAs, and prove its generalization ability. **(3)** Unlike prior studies that mainly focus on small-scale tasks, we conduct extensive numerical studies to validate the advancements of PALQO up to 40 qubits, providing valuable insights to further improve the optimization efficiency of VQAs at scale.

## 2 Preliminaries

In this section, we provide a concise overview of the basic concepts of quantum computing, variational quantum algorithms and physics-informed neural networks to set the stage for their integration in accelerating the quantum optimization process. Please refer to Appendix A for more details.

**Basics of Quantum Computing.** Quantum state, quantum circuit, and quantum measurement are three key components in quantum computing [43, 44]. In particular, an $n$-qubit quantum state is mathematically represented as a unit vector $\mathbf{u} \in \mathbb{C}^N$ in Hilbert space, where $\sum_{j=0}^{N-1} |\mathbf{u}_j|^2 = 1$, $N = 2^n$. Here we follow conventions to use Dirac notation to represent $\mathbf{u}$ and its transpose conjugate $\mathbf{u}^\dagger$ ,

i.e., $|\mathbf{u}\rangle$ and $\langle\mathbf{u}|$. For a quantum circuit, it serves as a computational model consisting of a sequence of quantum gates that describes operations on the given input state. The most widely used quantum gates are Pauli gates, i.e., $X = \left(\begin{smallmatrix} 0 & 1 \\ 1 & 0 \end{smallmatrix}\right)$, $Z = \left(\begin{smallmatrix} 1 & 0 \\ 0 & -1 \end{smallmatrix}\right)$, $Y = \left(\begin{smallmatrix} 0 & -i \\ i & 0 \end{smallmatrix}\right)$. According to the Solovay-Kitaev theorem [43], arbitrary operation can be approximated by a quantum circuit $U = \prod_j U_j$ where each gate $U_j$ is drawn from a finite universal gate set, such as $\{\text{CNOT}, H, S, R_z(\theta), R_x(\theta)\}$. Concretely, $\text{CNOT} = |0\rangle\langle 0| \otimes \mathbb{I} + |1\rangle\langle 1| \otimes X$, $H = 1/\sqrt{2}\left(\begin{smallmatrix} 1 & 1 \\ 1 & -1 \end{smallmatrix}\right)$, $R_x(\theta) = e^{-i\theta X}$, $R_z(\theta) = e^{-i\theta Z}$, $S = \sqrt{Z}$.

For quantum measurement, it is the process that collapses a quantum state into a definite classical outcome. In this study, we are interested in the expectation value of the measurement outcomes with a given observable $O$, a Hermitian operator, on quantum state $|\mathbf{u}\rangle$, i.e. $\langle\mathbf{u}|O|\mathbf{u}\rangle$. Suppose the observable presents as an $n$-qubit Hamiltonian that characterizes energy structure of the target quantum system in the form of $H = \sum_{j=1}^{N_H} c_j P_j$, where $P_j$ is a tensor product of Pauli matrices, i.e. $P_j \in \{\mathbb{I}, X, Y, Z\}^{\otimes n}$. To experimentally estimate $\langle\mathbf{u}|O|\mathbf{u}\rangle$ within error $\epsilon$, we typically perform $M \sim \mathcal{O}(1/\epsilon^2)$ repeated measurements for each $P_j$ on multiple copies of the state $|\mathbf{u}\rangle$ and get the outcomes $\{\hat{M}_{\mathbf{u}}^{j,k}\}_{k=1,...,M}$, then approximate the expectation value by statistical averaging $\langle\mathbf{u}|O|\mathbf{u}\rangle = 1/(MN_H) \sum_{j,k} c_j \hat{M}_{\mathbf{u}}^{j,k}$.

**Variational Quantum Algorithms.** Variational quantum algorithms (VQAs) algorithms designed for machine learning tasks are called quantum neural networks (QNNs) [41, 45–52], while those applied to many-body physics and quantum chemistry are typically known as variational quantum eigensolvers (VQEs) [53–60]. The primary objective of VQE is to optimize a parameterized state $|\psi(\boldsymbol{\theta})\rangle = U(\boldsymbol{\theta})|\phi\rangle$ to minimize the energy function $\mathcal{E}$ defined by a given Hamiltonian $H = \sum_{j=1}^{N_H} c_j P_j$. Mathematically, the energy function to be minimized in VQEs takes the form of

$$\min_{\boldsymbol{\theta}} \mathcal{E}(\boldsymbol{\theta}) = \langle\psi(\boldsymbol{\theta})|H|\psi(\boldsymbol{\theta})\rangle. \tag{1}$$

A common and widely adopted approach to complete this optimization problem is utilizing a gradient-based optimizer, like gradient descent, to iteratively adjust the parameters $\boldsymbol{\theta}$ according to the partial derivative $\partial_{\boldsymbol{\theta}}\mathcal{E}$. Because there is no-clone theorem and no backpropagation without exponential classical overhead in general VQEs [24], we need to perform the *parameter shift rule* without involving other quantum resource overhead, such as ancillary qubits to estimate the partial derivative [25]. Concretely, the calculation of the partial derivative with respect to $\boldsymbol{\theta}_i$ takes the form as

$$\frac{\partial \mathcal{E}}{\partial \boldsymbol{\theta}_i} = \frac{1}{2}\left[\mathcal{E}\left(\boldsymbol{\theta}_i + \frac{\pi}{2}\right) - \mathcal{E}\left(\boldsymbol{\theta}_i - \frac{\pi}{2}\right)\right] \approx \frac{1}{MN_H} \sum_{j,k} c_j \left[\hat{M}_{\psi_+}^{j,k} + \hat{M}_{\psi_-}^{j,k}\right], \tag{2}$$

where $\psi_+, \psi_-$ correspond to state $|\psi(\boldsymbol{\theta}_i + \pi/2)\rangle$ and $|\psi(\boldsymbol{\theta}_i - \pi/2)\rangle$, respectively.

While such a method provides a closed-form expression for gradient estimation without requiring additional qubits and can be extended to general VQEs, it necessitates evaluating $\mathcal{E}$ twice with shifted parameter values at the same position to estimate the gradient of a single parameter. Hence, suppose the dimension of $\boldsymbol{\theta}$ is $p$, it requires to perform $\mathcal{O}(pN_H/\epsilon^2)$ measurements to estimate the partial derivative $\partial_{\boldsymbol{\theta}}\mathcal{E}$, which becomes **computationally prohibitive** in large-scale tasks, especially for large molecules as whose $n$-qubits second-quantized Hamiltonian has roughly $N_H \sim \mathcal{O}(n^4)$ terms [53]. Therefore, it requires substantial resources for estimating updated parameters $\boldsymbol{\theta}$ during the optimization, which is considered as one major limitation of large-scale VQEs. Similarly, such a scalability issue also arises in QNNs, where the number of measurements required per iteration scales linearly with $p$ and the batch size.

**Physical-Informed Neural Network.** Physics-informed neural networks (PINNs) have become a promising learning-based tool in approximating the solution of partial differential equations (PDEs) [61–63]. With the advantages of computational efficiency for solving complex PDE, they have been widely employed in various practical scenarios such as fluid dynamics, battery degradation modeling, disease detection, and complex systems simulation [38, 64–67]. PINNs harness the core tool, automatic differentiation, of modern machine learning to efficiently enforce the physical constraints of the underlying PDE.

For a PDE problem, it can be generally written as $\mathcal{N}[u(\boldsymbol{x}, t)] = g(\boldsymbol{x}, t)$ where $\boldsymbol{x} \in \mathcal{D} \subset \mathbb{R}^d$ denotes variables, $\mathcal{N}$ represents the differential operators, $u(\boldsymbol{x}, t)$ stands for the solution, and $g(\boldsymbol{x}, t)$ refers to input or source function. The aim of PINNs is to build a neural network $f_{\boldsymbol{w}}$ with parameters $\boldsymbol{w}$ to approximate the true solution $u$. Hence, the loss function of PINN for solving a general PDE is based on *residuals*, including PDE residual and data residual. The PDE residual measures the difference

between the neural network solution and the true solution, expressed as

$$\mathcal{L}_P = \sum_j \left| \mathcal{N}\left[ f_{\boldsymbol{w}}\left( \boldsymbol{x}_p^{(j)}, t_p^{(j)} \right) \right] - g\left( \boldsymbol{x}_p^{(j)}, t_p^{(j)} \right) \right|^2, \quad \mathcal{L}_D = \sum_j \left| f_{\boldsymbol{w}}\left( \boldsymbol{x}_d^{(j)}, t_d^{(j)} \right) - u_d^{(j)} \right|^2. \quad (3)$$

Here, $\{\boldsymbol{x}_p^{(j)}, t_p^{(j)}\}_{j=1}^{N_p}$ used in $\mathcal{L}_P$ are selected collocation points for enforcing PDE structure. For $\mathcal{L}_D$, the dataset $\{\boldsymbol{x}_d^{(j)}, t_d^{(j)}, u_d^{(j)}\}_{j=1}^{N_d}$ with $u_d^{(j)} = u(\boldsymbol{x}_d^{(j)}, t_d^{(j)})$ denote the training data on $u(\boldsymbol{x}, t)$ [37]. Thus, the total loss is putting all residuals together, i.e. $\mathcal{L} = \mathcal{L}_P + \mathcal{L}_D$. By embedding physical principles into the learning process, PINNs serve as a versatile tool that only requires a small amount of data to tackle the computationally complex problem.

## 2.1 Related Works

Prior literature related to improving the optimization efficiency of VQAs can be classified into three main classes, i.e., *measurement grouping*, *initializer design*, and *prediction of training dynamics*. Since the first two classes are complementary to PALQO, we defer the explanations to Appendix B.

The third class aims to harness learning models to approximate the training process. Some works inspired by meta-learning utilize the recurrent neural network to learn a sequential update rule in a heuristic manner [34, 35]. Nevertheless, the memory bottleneck and training instability of the recurrent neural network would lead to it being underwhelming [68]. Recent work proposed QuACK, involving linear dynamics approximation and nonlinear neural embedding, to accelerate the optimization [36]. However, the prediction phase requires estimating the energy loss of each step to find the optimal parameters, which is not friendly for large-scale problems. To overcome these limitations, the proposed PALQO uniquely approximates VQA training dynamics using a nonlinear PDE, embedding the dynamical laws directly into the learning process. In this way, it offers *deeper physical insight* and achieves *superior performance* through principled model-guided optimization.

## 3 PALQO: physics-informed model for accelerating quantum optimization

In this section, we first formally define the problem of learning the training dynamics of VQAs as nonlinear PDE problems in Sec. 3.1. Then, in Sec. 3.2, we introduce PALQO to solve this PDE via a tailored PINN-based model, where the optimized solutions correspond to the optimization trajectory of VQAs, followed by a generalization error analysis.

### 3.1 Reformulating the optimization of VQA as a PDE problem

Recall the optimization of VQEs in Sec. 2. As it is costly in querying $\boldsymbol{\theta}^{(t)}$ of each step $t$, it is demanded to develop a protocol that only learns from a few trajectory data $\{\boldsymbol{\theta}^{(t)}\}_{t=1}^{\mathcal{T}}$ to classically predict future steps, thereby avoiding prohibitive resource costs without compromising accuracy. To achieve this goal, we start by revisiting the gradient descent dynamics of VQEs. The updating rules of parameters $\boldsymbol{\theta}$ with learning rate $\eta$ at step $t$ is given by $\delta\boldsymbol{\theta} = \boldsymbol{\theta}^{(t+1)} - \boldsymbol{\theta}^{(t)} = -\eta \partial\mathcal{E}(\boldsymbol{\theta})/\partial\boldsymbol{\theta}$, where $\partial\mathcal{E}(\boldsymbol{\theta})/\partial\boldsymbol{\theta}$ is estimated through phase shift rule shown in Eq. (2). Suppose $\eta$ is infinitesimally small, the following ordinary differential equation, a.k.a., gradient flow, characterizes how parameters change in continuous time, i.e.,

$$\frac{\partial\boldsymbol{\theta}}{\partial t} = -\frac{\partial\mathcal{E}}{\partial\boldsymbol{\theta}}. \quad (4)$$

Besides, we can similarly define the dynamics of $\mathcal{E}$ in a general form under Taylor expansion as

$$\frac{\partial\mathcal{E}}{\partial t} = -\sum_i \left( \frac{\partial\mathcal{E}}{\partial\boldsymbol{\theta}_i} \right)^2 + \frac{\eta}{2} \sum_{j,k} \frac{\partial^2\mathcal{E}}{\partial\boldsymbol{\theta}_j \partial\boldsymbol{\theta}_k} \frac{\partial\mathcal{E}}{\partial\boldsymbol{\theta}_j} \frac{\partial\mathcal{E}}{\partial\boldsymbol{\theta}_k} + \mathcal{O}(\eta^2), \quad (5)$$

where the first term of RHS of Eq. (5) is termed as quantum neural tangent Kernel (QNTK) [39, 40, 69], which captures the sensitivity of outputs to parameter changes, shaping how the gradient flow evolves in parameter space. The second term involves the Hessian matrix of $\mathcal{E}$ that reflects the local curvature of the cost function in the optimization landscape. Thus, tackling the problem of learning optimization trajectory can be recast to solve PDEs presented in Eqs. (4) and (5). This reformulation

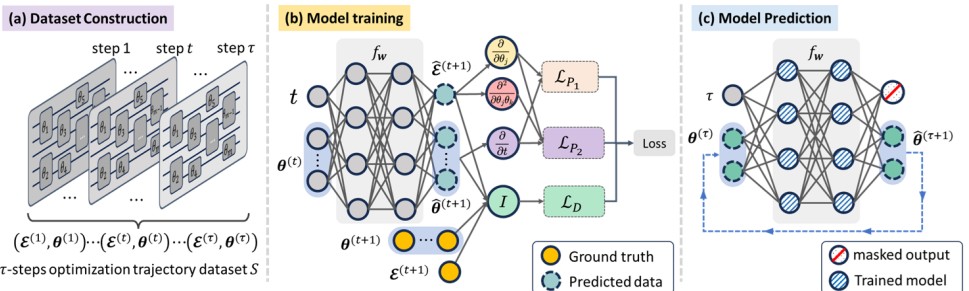

Figure 1: **Framework of PALQO**. (a) Dataset construction: the tunable angles $\boldsymbol{\theta}$ and corresponding loss $\mathcal{E}$ from a VQE over $\tau$ optimization steps are collected, forming a sequential training dataset that captures the optimization trajectory over time. (b) Model training: A PINN $f_{\boldsymbol{w}}$ is trained by minimizing the loss $\mathcal{L} = \lambda_{P_1}\mathcal{L}_{P_1} + \lambda_{P_2}\mathcal{L}_{P_2} + \lambda_D\mathcal{L}_D$. (c) Prediction: starting with the last step $\boldsymbol{\theta}^{(\tau)}$, the trained $f_{\boldsymbol{w}}$ is used to recurrently predict parameters, mimicking the optimization process without access to the quantum device.

provides a powerful framework and allows us to leverage a rich set of tools developed for PDEs to understand the underlying behavior of the optimization process in large-scale VQEs.

**Remark**. In Appendix C.1, we elucidate the derivation of Eq. (5) and its relation to QNTK. Besides, the above reformulation can be effectively extended to broader VQAs such as QNNs.

### 3.2   Implementation of PALQO

In light of the above reformulation, we propose PALQO based on PINN introduced in Eq. (3) to learn the optimization trajectory of large-scale VQEs. Conceptually, once PALQO attains a low training error, it can predict the optimization path, which substantially reduces the measurement cost as considered in large-scale VQEs. As such, it enhances scalability and resource efficiency by minimizing the need for extensive quantum circuit evaluations while maintaining high fidelity in modeling complex quantum optimization.

An overview of our protocol is depicted in Fig. 1, which consists of consists of three stages: *Dataset construction, Model training, and Model prediction*. In Fig. 1 (a), it starts by generating a dataset $\mathcal{S} = \{\boldsymbol{\theta}^{(t)}, \mathcal{E}^{(t)}\}_{t=1}^{\tau}$ corresponding to trajectory data consisting of $\boldsymbol{\theta}$ and energy loss $\mathcal{E}$ over $\tau$ optimization steps collected from a quantum device. Then, as shown in Fig. 1 (b), PALQO utilizes the collected $\mathcal{S}$ to train a PINN-based model $f_{\boldsymbol{w}}$ to capture the underlying optimization dynamics. Given the trained $f_{\boldsymbol{w}}$, as shown in Fig. 1 (c), it can recursively predict the parameters $\boldsymbol{\theta}$ of the future steps until it convergence, i.e. $|\boldsymbol{\theta}^{(t)} - \boldsymbol{\theta}^{(t+1)}| + |\mathcal{E}^{(t)} - \mathcal{E}^{(t+1)}| \leq \varsigma$ with $\varsigma$ being a small constant.

**Dataset construction.** To mimic the dynamic of VQE optimization, we need first perform $\tau$ steps optimization via gradient descent to collect a sequential trajectory as the training data, including loss $\{\mathcal{E}^{(1)}, \mathcal{E}^{(2)}, \ldots, \mathcal{E}^{(\tau)}\}$ and $\{\boldsymbol{\theta}^{(1)}, \boldsymbol{\theta}^{(2)}, \ldots, \boldsymbol{\theta}^{(\tau)}\}$ where $\boldsymbol{\theta}^{(j)} = (\theta_1^{(j)}, \cdots, \theta_p^{(j)}) \in \mathbb{R}^p$ is the parameters at $j$-th epoch. Notably, assume a VQE with a total $T$ optimization steps, PALQO only needs $\tau \ll T$ steps to construct the training dataset $\mathcal{S}$. This is because the PINN-based model leverages strong inductive biases from the dynamical laws, such that it does not need to infer fundamental principles from scratch, allowing the model to generalize well from limited data.

**Model training.** Once the dataset is collected, PALQO employs a deep neural network $f_{\boldsymbol{w}}$ that takes $\{(t, \boldsymbol{\theta}^{(t)})\}_{t=1}^{\tau}$ comprising $t$-th time step and trajectory data $\boldsymbol{\theta}^{(t)}$ as the input, and predict the loss and parameters for the $(t+1)$-th step, i.e.,$(\mathcal{E}^{(t+1)}, \boldsymbol{\theta}^{(t+1)})$. Refer to Appendix C for the details about the neural architecture of $f_{\boldsymbol{w}}$. Denote the prediction of $f_{\boldsymbol{w}}$ for the $t$-th step as $(\hat{\boldsymbol{\theta}}^{(t)}, \hat{\mathcal{E}}^{(t)})$. Through leveraging the automatic differentiation capabilities of neural networks, the derivatives of outputs with respect to inputs, i.e., $\partial\hat{\boldsymbol{\theta}}/\partial t$ and $\partial\hat{\mathcal{E}}/\partial\boldsymbol{\theta}$, can be efficiently computed on classical devices. This efficiency enables the direct incorporation of dynamical law constraints, as described in Eqs. (4) and (5), into the loss function. In this regard, we devise the loss function of PALQO as

$$\mathcal{L} = \lambda_D\mathcal{L}_D + \lambda_{P_1}\mathcal{L}_{P_1} + \lambda_{P_2}\mathcal{L}_{P_2}, \tag{6}$$

where $\{\lambda_D, \lambda_{P_1}, \lambda_{P_2}\}$ denote hyperparameters to balance the data-driven loss $\mathcal{L}_D$ and two PDE residual losses $\mathcal{L}_{P_1}$ and $\mathcal{L}_{P_2}$. In particular, the data residual loss is defined as $\mathcal{L}_D = \sum_{t=1}^{\tau}(\mathcal{E}^{(t)} -$

$\hat{\mathcal{E}}^{(t)})^2 + \sum_{t=1}^{\tau}(\boldsymbol{\theta}^{(t)} - \hat{\boldsymbol{\theta}}^{(t)})^2$, aiming to capture the temporal changes of $\mathcal{E}$ and $\boldsymbol{\theta}$ among $\tau$ steps. Meanwhile, the two PDE residual losses aim to enforce the PINN to capture the evolution of VQE optimization dynamics through the underlying derivative structure of the loss landscape. Following Eq. (4), the explicit form of the first PDE loss is $\mathcal{L}_{P_1} = \sum_{t=1}^{\tau}(\sum_{j=1}^{p}(\partial\hat{\boldsymbol{\theta}}_j^{(t)}/\partial t + \partial\hat{\mathcal{E}}^{(t)}/\partial\boldsymbol{\theta}_j^{(t)}))^2$. By focusing on the first two orders of the derivatives in Eq. (5), the second PDE loss yields

$$\mathcal{L}_{P_2} = \sum_{t=1}^{\tau}\Big(\frac{\partial\hat{\mathcal{E}}^{(t)}}{\partial t} + \sum_{j=1}^{p}\Big(\frac{\partial\hat{\mathcal{E}}^{(t)}}{\partial\boldsymbol{\theta}_j^{(t)}}\Big)^2 - \frac{\eta}{2}\sum_{j,k=1}^{p}\frac{\partial^2\hat{\mathcal{E}}^{(t)}}{\partial\boldsymbol{\theta}_j^{(t)}\partial\boldsymbol{\theta}_k^{(t)}}\frac{\partial\hat{\mathcal{E}}^{(t)}}{\partial\boldsymbol{\theta}_j^{(t)}}\frac{\partial\hat{\mathcal{E}}^{(t)}}{\partial\boldsymbol{\theta}_k^{(t)}}\Big)^2. \tag{7}$$

The model $f_{\boldsymbol{w}}$ is optimized by minimizing $\mathcal{L}$ in Eq. (6) via a gradient-based optimizer Adam [70].

**Model prediction.** As the trained $f_{\boldsymbol{w}}$ can not only approximate the solution of the underlying PDE but also capture temporal dependencies of the trajectory data, we are able to recurrently predict the upcoming updates of the $\boldsymbol{\theta}$. As shown in Fig. 1 (c), by passing $(\tau, \boldsymbol{\theta}^{(\tau)})$ through the trained $f_{\boldsymbol{w}}$ and masking the $\hat{\mathcal{E}}^{(\tau)}$ node, we can obtain the predicted data $\hat{\boldsymbol{\theta}}^{(\tau+1)}$, and then fed $(\tau + 1, \hat{\boldsymbol{\theta}}^{(\tau+1)})$ as the input back to the network to make the followed prediction. It is worth noting that directly making long-term forecasts to reach the optimal solution of the target problem is exceedingly challenging. To that end, we employ the non-overlapping sliding windows to enhance the network for long-term prediction. More details on the prediction process can be found in Appendix C.

**Remark.** The second PDE residual loss $\mathcal{L}_{P_2}$ can be extended to arbitrary higher orders. Empirical results indicate that a second-order approximation offers a sufficient balance between accuracy and computational cost for modeling the optimization dynamics of the VQEs studied herein. Moreover, while we mainly focus on learning the training process of VQE, our model can be efficiently extended to more general tasks such as quantum machine learning [19, 20, 22], quantum simulation [71–73], and quantum optimization [74, 75] by slightly modifying the Eqs. (4) and (5). See Appendix F for details.

Building upon prior work on the error analysis of PINNs [76], we conduct the analysis of the Lipschitz constant bound for PALQO and derive a corollary to establish a generalization bound for PALQO applied to VQEs. An informal statement of the derived generalization bound is provided below, where the formal statement and the related proof are deferred to Appendix D.

**Corollary 3.1.** *(Informal) When utilizing PALQO, whose PINN is constructed by a $L$ layer tanh neural network with most $W$ width of each layer and trained over $\tau$ data samples, to approximate the solution of PDE that describes training dynamics of a VQE with $p$ tunable parameters $\boldsymbol{\theta}$, with probability at least $1 - \gamma$, its the generalization error is upper bounded by*

$$\tilde{\mathcal{O}}\left(\sqrt{\frac{pL^2W^2}{\tau}\left(\ln\left(\frac{p}{\epsilon}\right) + \ln\left(\frac{1}{\gamma}\right)\right)}\right). \tag{8}$$

The achieved results indicate that for any $\epsilon > 0$, the number of training examples scales at most polynomially in $p$, $L$, and $W$ is sufficient to guarantee a well-bounded generalization error. This warrants the applicability of PALQO in large-scale scenarios.

## 4 Experiments

To evaluate the practical performance of the proposed PALQO framework, we apply it to two representative quantum applications: finding ground state energy of many-body quantum system and molecules in quantum chemistry, which have broad applicability in understanding many-body physical phase transitions [77–79] and simulation of complex electronic structures of molecular systems in drug design and discovery [15, 80, 81]. The concrete settings are elucidated below.

**Many-body quantum system.** A many-body quantum system consists of interacting quantum particles whose collective behavior and correlations lead to complex phenomena beyond single-particle descriptions. Here, we consider three typical many-body quantum systems. 1) **Transverse-field Ising model (TFIM)** describes spin particles on a lattice interacting via nearest-neighbor coupling and subject to a transverse magnetic field, whose Hamiltonian is typically in a form of $H_{\text{TFIM}} = -J\sum_j Z_j \otimes Z_{j+1} - h\sum_j X_j$, where $Z_j$ and $X_j$ refer to the Pauli matrices $Z$ and $X$ applied on the $j$-th qubit, respectively. 2) **Quantum Heisenberg model** also describes the spin

particles on a lattice, but spin-spin interactions occur along all spatial directions. Its Hamiltonian can be represented as $H_{QH} = -1/2 \sum_j (J_x X_j X_{j+1} + J_y Y_j Y_{j+1} + J_z Z_j Z_{j+1} + h Z_j)$. 3) **Bond-alternating XXZ model** is an anisotropic variant of the Heisenberg model with unequal coupling strengths in the transverse and longitudinal directions. The Hamiltonian is given by $H_{XXZ} = \sum_{j=odd} J(X_j X_{j+1} + Y_j Y_{j+1} + \delta Z_j Z_{j+1}) + \sum_{j=even} J'(X_j X_{j+1} + Y_j Y_{j+1} + \delta Z_j Z_{j+1})$. The coefficients $J, h, J_x, J_y, J_z, J', \delta$ within the explored Hamiltonians represent the coupling strength that determines the ground state properties and phase transitions.

**Quantum chemistry.** The Hamiltonian of a molecule describes the total energy of its electrons and nuclei and serves as the fundamental operator for determining the molecule's electronic structure in quantum chemistry. The general form of the molecule Hamiltonian can be presented as $H = \sum_j h_j P_j$ where $P_j$ represents tensor products of Pauli matrices, and $h_j$ are the associated real coefficients. Here, we select a widely studied molecule—LiH, and a relatively large and challenging $BeH_2$ molecule as the target molecules [8, 53, 82]. We generate these molecule Hamiltonians with Openfermion [83]. Refer to Appendix E for more details about the molecule experiments.

## 4.1 Experimental Setup

We employ three standard ansatzes, i.e., hardware-efficient ansatz (HEA) [84–86], Hamiltonian variable ansatz (HVA) [87–89], unitary coupled cluster with single and double excitations ansatz (UCCSD) [90–92], to implement VQE for the different Hamiltonians mentioned above. These ansatzes adopt a layered architecture. Refer to Appendix E.2 for the implementation of these ansatzes. For all ansatzes, their initial parameters $\boldsymbol{\theta}^{(0)}$ are uniformly sampled from $[0, 1]$, following the strategy adopted in QuACK [36]. The gradient descent is set as the default optimizer.

For implementing PALQO, we randomly initialize the parameters $\boldsymbol{w}$ of the neural network $f_{\boldsymbol{w}}$ from $[-1, 1]$ and employ the Adam as the optimizer, where the architectures of $f_{\boldsymbol{w}}$ are listed in Appendix C.2. To improve the training stability and convergence, we utilize a linear decay strategy to adaptively adjust the learning rate during training. Besides, the weight hyperparameters in loss function $\lambda_{P_1}, \lambda_{P_2}, \lambda_D$, are set as $1.0, 1.0$ and $10^{-4}$, respectively.

**Benchmark models.** To show the outperformance of PALQO against the state-of-the-art methods, we introduce the following baseline and benchmark. First, we use a *vanilla VQE* as the baseline since it provides a well-established reference point to evaluate improvements in accuracy, convergence, and efficiency. Second, we select a *LSTM-based model* [34] as a benchmark since it provides a strong reference for evaluating methods in modeling the temporal dependencies and iterative dynamics of optimization trajectories. Third, we pick *QuACK* [36] as another benchmark as it represents an advanced approach that learns surrogate dynamics of VQAs by embedding Koopman operator-based linear representations into nonlinear neural networks. The implementation of these reference models is deferred to Appendix E.3.

**Evaluation metrics.** To quantify the performance of PALQO in accuracy and efficiency, we consider the following metrics, 1) **Accuracy.** we define the accuracy as how close the estimated energy $\hat{E}$ is to a given target energy $E$ of a quantum system, i.e. $\Delta E = |\hat{E} - E|$. 2) **Efficiency.** we define the speedup ratio as $\alpha = \mathcal{I}_P / \mathcal{I}_V$, where $\mathcal{I}_P$ and $\mathcal{I}_V$ refer to the number of iterations required by the baseline method (vanilla VQE) and PALQO or other benchmark models, respectively, to achieve an acceptable accuracy $a$. Specifically, we set $a \leq 10^{-3}$.

## 4.2 Experimental Results

We next evaluate the performance of PALQO and other reference models when applied to the aforementioned Hamiltonians under different settings.

**PALQO significantly reduces the measurement overhead.** Here, we utilize the number of measurements incurred during the optimization as a quantum resource measure to explore the performance of PALQO and the other benchmark models when applied to 20 qubits TFIM with HEA, 20 qubits Heisenberg model with HVA, and 14 qubits $BeH_2$ with UCCSD ansatz, The number of parameters for each case are $120, 180, 90$, respectively. As shown in Tab. 1, PALQO achieves significant quantum resource efficiency in aforementioned tasks, with around 90% average reduction in measurement overhead while preserving $\Delta E$ around $10^{-3}$.

These substantial savings stem from two key factors: 1) compared to the vanilla VQE, PALQO leverages PINN to predict parameter updates, thereby reducing reliance on frequent quantum measurements; 2) the rapid convergence on the classical side enables further reduction in quantum resource expenditure.

Table 1: The number of quantum measurement shots ($\times 10^8$) required for TFIM, Heisenberg model, and BeH$_2$.

| SYSTEM SIZE | $H_{\text{TFIM}} = 20$ | $H_{\text{HQ}} = 20$ | $H_{\text{BeH}_2} = 14$ |
|---|---|---|---|
| VANILLA VQE | 10.97 | 21.66 | 464.3 |
| LSTM | 3.126 | 14.49 | 312.4 |
| QUACK | 5.217 | 14.15 | 461.8 |
| PALQO | **1.535** | **5.749** | **28.01** |

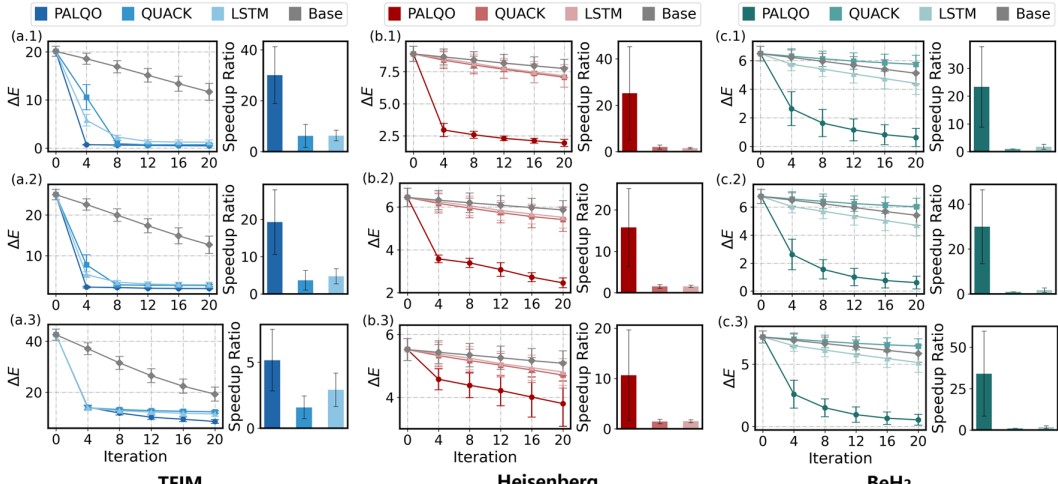

**TFIM**          **Heisenberg**          **BeH₂**

Figure 2: Performance comparison between PALQO and the reference models in 20 qubits TFIM with HEA, 20 qubits Heisenberg model with HVA, and 14 qubits BeH2 with UCCSD ansatz. Each subplot comprises a $\Delta E$ curve over iterations performed on a quantum device, along with a bar chart depicting the speedup ratios achieved by PALQO and competing models. The left column illustrates results for TFIM with $J/h = \{2, 1, 0.5\}$. The central column shows results for Heisenberg model with $J_x = J_y = h = 1, J_z = \{0.5, 1, 2\}$. The right column displays the model performance on BeH2 with the bond length $b = \{1.3, 1.4, 1.5\}$.

**PALQO outperforms benchmark models in accuracy and efficiency.** The performance comparisons of PALQO on 20 qubits TFIM with HEA, 20 qubits Heisenberg model with HVA, and 14 qubits BeH2 with UCCSD ansatz for varying structural parameters are presented in Fig. 2. In particular, we observed that PALQO consistently outperformed, up to 30x speedup and lower $\Delta E = |\hat{E} - E|$ around $10^{-3}$, like the case of TFIM with $J/h = 2$ and HEA ansatz, compared to the other evaluated approaches. Furthermore, as the PALOQ predicts the future optimization steps on classical hardware, it exhibited a faster rate of convergence, achieving a substantial reduction in $\Delta E$ within fewer iterations performed on a quantum device, compared to the baseline methods. Although the speedup ratio of PALQO has a relatively large variance, its minimum value remains comparable to the average performance of the other approaches. Refer to Appendix F for the results of XXZ model and LiH.

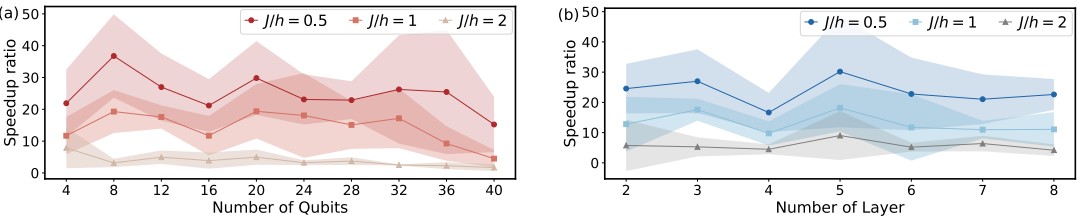

Figure 3: Scalability analysis of PALQO on TFIMs. (a) The speedup ratio achieved by PALQO in modeling VQE training dynamics with a fixed HEA, ranging from 4 to 40 qubits. (b) The speedup ratio of PALQO with a fixed system size of 12 qubits, assessed under increasing HEA ansatz layers from 2 to 8.

**Scalability of PALQO.** We next investigate the scalability of PALQO on the TFIM with HEA, examining its performance in increasing system sizes (from $n = 4$ to $n = 40$) and the number of ansatz layers (from 2 to 8). In Fig. 3, the results reveal that the speedup of PALQO is contingent upon the specific system configuration. Nevertheless, as shown in Fig. 3 (a), while the speedup ratio fluctuates with the number of qubits varying, it still achieves up to 30x speedup when $J/h = 0.5$. The lower speedup at $J/h = 2$ is due to a smaller energy gap between the ground and first excited states, making the optimization more challenging. Similar behavior also appears in Fig. 3 (b). This suggests that the performance benefits of PALOQ are maintained as the computational demands grow, indicating its potential for large-scale quantum optimization.

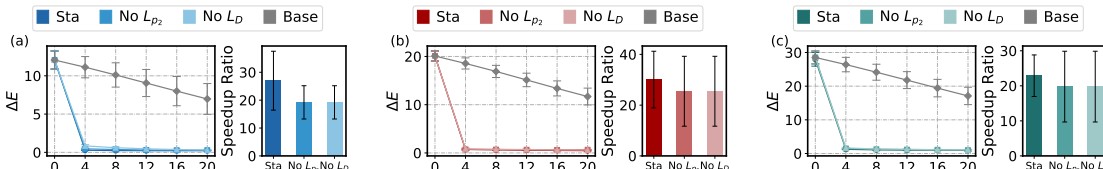

Figure 4: Ablation study on the loss function configuration in PALQO. The three panels show the performance on TFIM with $n = 12, 20, 28$ qubits, evaluating the impact of the components $\mathcal{L}_D$ and $\mathcal{L}_{P_2}$ in $\mathcal{L}$.

**Ablation studies on loss function.** We further evaluate the performance of PALOQ against variants where specific loss terms in Eq. (6) are removed in the task of TFIM with HEA ansatz. Specifically, we carried out separate ablation studies on the PDE residual and data residual components of the loss function shown in Fig. 4. We noticed that while both the PDE and data residual positively influence model performance, their contributions are not essential. These findings suggest that adopting low-order approximations during the construction of PALQO enables the retention of satisfactory speedup while simultaneously reducing the complexity of downstream model training and preventing the degradation of higher-order derivative information.

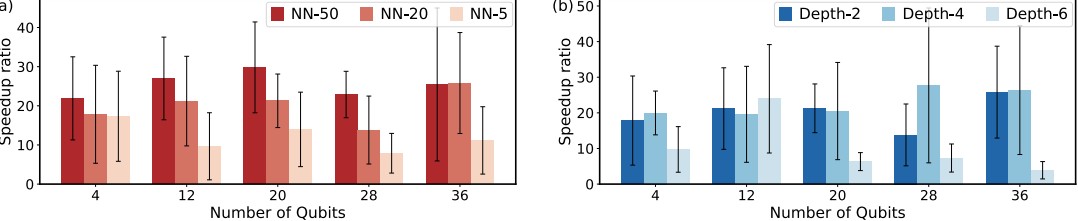

Figure 5: Performance of PALQO evaluated on the TFIM under different neural network architecture configurations. (a) Results obtained using 2-layer neural networks with varying hidden layer widths of $W = \{50p, 20p, 5p\}$, where $p$ denotes the number of parameters $\boldsymbol{\theta}$. (b) Results with a fixed hidden layer width of $20p$, varying the number of hidden layers from 2 to 6.

**Performance on varying the size of PALQO.** To investigate the influence of varying neural network sizes (width $W$ and depth $L$) within PALQO on its performance, we conducted tests on the TFIM with HEA ansatz and $p = 6n$ parameters, where $n$ is the system size varying from 4 to 36. In Fig. 5 (a), we observed that an increase in the width of the hidden layers leads to a corresponding improvement in speedup ratio. However, an inverse phenomenon occurs in Fig. 5 (b), further increasing the neural network depth does not effectively enhance PALQO's performance, which may be related to the vanishing gradient phenomenon, where higher-order derivative information tends to diminish as the neural network becomes deeper [93]. These observations provide guidance for the neural network design in PALQO, indicating that increasing hidden layer width should be prioritized.

**The impact of the number of training samples on model prediction.** We performed experiments on 12-qubit TFIMs with HEA ansatz using various sample sizes to explore how the number of training samples affects the performance. In Fig. 6, the results validate that PALQO can achieve satisfactory performance even with a limited number of

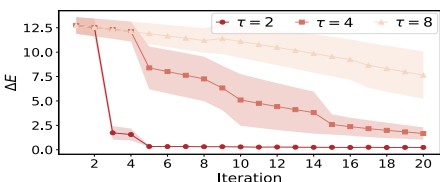

Figure 6: Performance comparison of PALQO when trained with 2, 4, and 8 data samples.

training samples. Such data efficiency
arises from the direct integration of the
physical constraints imposed by the governing PDE into the loss function of the neural network, a
core characteristic of PINNs [63].

## 5    Conclusion

In this study, we devised PALQO towards optimizing large-scale VQAs given restrictive quantum
resources. In contrast to previous studies, we derive PALQO from reformulating the training dynamics
as a nonlinear PDE and using PINN to approximate the solution, and also provide a generalization
analysis. Extensive numerical experiments up to 40 qubits validate the effectiveness of PALQO.
Although it is still uncertain whether PALQO can scale to the regime where quantum hardware
decisively outperforms classical methods, its results at the currently accessible scale are highly
encouraging.

**Limitations and future works**    PALQO reduces the need for repeated quantum gradient evaluations
by learning the optimization path classically. While this lowers the number of quantum queries
compared to vanilla VQE, it may diminish some quantum advantages. Besides, mitigating the high
variance in speedup ratios is crucial for achieving more stable and reliable performance. One future
research direction is to incorporate adaptive strategies and variance reduction techniques to achieve
this goal and further unlock its potential.

## 6    Acknowledge

We thank Yusen Wu for the helpful discussions and the anonymous reviewers for their constructive
feedback and valuable suggestions. This work was supported by Innovation Program for Quantum
Science and Technology (Grant No. 2023ZD0300200), the National Natural Science Foundation of
China NSAF (Grant No. U2330201 and No. 92265208), Sichuan Science and Technology Program
(Grant No. 2025YFHZ0336). Y. D. acknowledges the support from the SUG grant of NTU. The
numerical experiments in this work were supported by the High-Performance Computing Platform of
Peking University.

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
