# OpenReview forum: "PALQO: Physics-informed model for Accelerating  Large-scale Quantum Optimization"
_NeurIPS.cc/2025/Conference — NeurIPS 2025 poster_

### Official Review · Reviewer_2QJF · 2025-06-04

**Clarity:** 3
**Significance:** 2
**Originality:** 3
**Rating:** 4
**Confidence:** 4

**Summary:**

This work explores the integration of Physics-Informed Neural Networks (PINNs) with Variational Quantum Eigensolvers (VQEs), as proposed in the PALQO framework. The key idea is to enhance the VQEs by reducing the number of measurements required by the parameter shift rule by predicting parameter values using PINNs trained and run on classical computers. PALQO achieves up to a 30x speedup compared to conventional methods and reduces quantum resource costs by up to 90% for quantum workloads involving up to 40 qubits.

**Questions:**

My fundamental question is about the quantum advantage retained with such a proposal.

(1) If the classical PINN model proposed by PALQO has to be scaled with VQE, what exactly is the quantum advantage?

(2) Would VQE retain any quantum advantage?

(3) How does the size of the PINN model scale with the size of the VQE?

(4) Is it exponential scaling?

(5) If not, what is the scaling required?

**Ethical Concerns:**

["NO or VERY MINOR ethics concerns only"]

**Final Justification:**

I have revised my score positively after an extensive discussion with the authors.

**Limitations:**

The paper includes one sentence about the limitations of this work in the conclusion, which discusses a potential future research direction of this work could be the reduce the high variance in speedup ratios of PALQO as the number of qubits and layers is varied. Other limitations discussed above are not mentioned in the work.

On the other hand, the work does not have any direct negative societal impact implications.

**Paper Formatting Concerns:**

The manuscript does not have any noticeable formatting deviations or issues.

**Quality:**

2

**Strengths And Weaknesses:**

Strengths:

VQE currently requires a prohibitive number of measurements. Reduction in the number of measurements, as facilitated by PALQO, could be very beneficial.

PALQO's PINN can achieve good performance even with a very limited number of training samples.

Weaknesses:

My fundamental question concerns the extent to which quantum advantage is preserved in the proposed approach. Specifically, if the classical PINN model used in PALQO must be scaled in conjunction with VQE, it raises the issue of what precise quantum advantage is actually obtained. The idea behind VQE is to only use classical resources for optimization, but if parameters also get resolved classically -- with the PINN essentially simulating the parameter shift rule -- then what is the need for a quantum VQE?

A related concern is whether VQE itself retains any inherent quantum advantage in this context. Furthermore, it is important to understand how the size of the PINN model scales relative to the size of the VQE. Is the scaling behavior exponential, or if not, what form of scaling is required to maintain efficacy?

There is a very high variance in speedup ratios of PALQO, which adversely affects its stability and reliability, especially as the number of qubits and number of ansatz layers (i.e., the size of the VQE) is varied. It is not exactly clear why this is happening.

---

> ### Author Rebuttal · Authors · 2025-07-29
>
> We sincerely thank Reviewer 2QJF for the constructive feedback, which has been instrumental in improving the quality of our submission. To ensure clarity and better align with the points raised under “Weaknesses,” we have marked the **Weakness** as **W1,W2,W3**, the **Questions** as **Q1–Q5** and **Limitation** as **L1** accordingly. We hope our detailed responses offer clear clarification and assist in the continued evaluation of our work.
>
> **[W1, Q1 & Q2]: The reviewer questions whether PALQO truly preserves quantum advantage, noting that if the classical PINN scales with VQE and simulates parameter updates, it may undermine the need for a quantum VQE.**
>
> **Response:** We thank the reviewer for raising the important point regarding the potential impact of our method on quantum advantage.
>
> The VQE remains a key focus in the field, as it currently lacks provable advantages on practical tasks but holds significant potential for near-term quantum computers [1,2]. While a definitive demonstration of quantum advantage is still on the horizon, recent there are still various effort to highlight the potential and progress of VQE [3,4].  Besides, recent work [5] shows that under certain type of ansatz and input state, there is the evidence of super-polynomial quantum speedups in quantum chemistry are theoretically achievable on near-term hardware [5].
>
>
> *$\bullet$ $\underline{\text{Preserves quantum advantage.}}$*
> We would like to clarify that PALQO does **not operate independently** of the VQE framework, nor does it diminish or negate the potential for quantum advantage. As explained in our response to **W2**, the PINN serves as a classical tool to analyze the training dynamics, but it does not replace the core quantum components that give VQE its power, such as the problem-agnostic ansatz and quantum state preparation.
>
> The PALQO framework is broadly applicable to various quantum algorithms, including the Quantum Approximate Optimization Algorithm (QAOA). While methods like PALQO aim to predict model parameters, QAOA is still believed to offer quantum advantage for solving certain classically hard problems—such as the Low Autocorrelation Binary Sequences (LABS) problem [6,7].
>
> *$\bullet$ $\underline{\text{Need for a quantum VQE.}}$* Yes, it requires quantum VQE. The core misconception is that the PINN "simulates" the VQE in a way that replaces it; instead, it only models the high-level dynamics of the VQE's classical parameters, a task for which it is entirely dependent on the quantum processor. The crux of the matter lies in the calculation of the energy gradient, which dictates the optimization path.
>
> **[W2,Q3-Q5]: A related concern is whether VQE retains any inherent quantum advantage in this context. Additionally, it is important to understand how the size of the PINN model scales relative to that of the VQE—is the scaling exponential, or if not, what kind of scaling is necessary to maintain its effectiveness?**
>
> **Response:** We thank the reviewer for the insightful comment.
>
> *$\bullet$ $\underline{\text{Retain inherent quantum advantage.}}$* The VQE retains its inherent quantum advantage in this context because the PINN operates purely on the classical side of the hybrid algorithm. The PINN does not attempt to simulate this complex quantum process; instead, it simply provides a powerful classical framework to learn and predict the optimization process by leveraging the gradient data generated by the VQE.
>
> *$\bullet$ $\underline{\text{The scale of PALQO.}}$*
> We would like to clarify that the PINN used to model the training dynamics of VQE avoids exponential scaling, as it operates in the space of classical parameters rather than the exponentially large quantum Hilbert space.
>
> To further support the above claims, we conducted scaling tests of PALQO on the transverse field Ising model (TFIM) using a hardware-efficient ansatz (HEA) across various system sizes, ranging from 4 to 36 qubits. The number of paramters $p$ scales linearly with the system size $n$, i.e. $p\sim O(n)$.
> In the following table, we present the speedup ratio of PALQO (with PINN) over vanilla VQE when converging to the ground state, under varying PINN width and depth. Specifically, we linearly increased the PINN width (from 5𝑝 to 50𝑝) and depth (from 2 to 6).
> In the left three columns of the figure, we fixed the depth at 2 and varied the width. We observed that even with linear increases in width, PINN consistently delivered stable speedup as the system size increased. Moreover, the speedup ratio steadily improved with wider networks. In the right three columns, we fixed the width at 20𝑝 and varied the depth. Again, PINN maintained stable acceleration across all tested depths.
> These scaling results suggest that PINN achieves strong performance with only linear increases in width and depth, avoiding the need for exponential scaling.
>
> |      |       | Width(l=2)|       |       | Depth(w=20p) |       |
> |------|-------|-------|-------|-------|-------|-------|
> |      | 5p    | 20p   | 50p   | 2     | 4     | 6     |
> | n=4  | 17±8  | 18±11 | 22±9  | 18±12 | 20±7  | 9±8   |
> | n=12 | 9±8   | 22±11 | 26±12 | 21±10 | 19±11 | 24±16 |
> | n=20 | 14±11 | 21±7  | 30±12 | 22±7  | 21±15 | 8±4   |
> | n=28 | 8±6   | 14±9  | 23±7  | 13±10 | 26±15 | 9±6   |
> | n=36 | 11±9  | 26±13 | 25±16 | 25±15 | 26±18 | 6±3   |
>
>
> **W3: There is a very high variance in speedup ratios of PALQO, which adversely affects its stability and reliability, especially as the number of qubits and number of ansatz layers (i.e., the size of the VQE) is varied. It is not exactly clear why this is happening.**
>
> **Response:** The observed high variance in the speedup ratio may be attributed to the following factors:
> 1. Like vanilla VQE, PALQO is sensitive to the choice of initialization. Because the VQE energy landscape is highly non-convex with many local minima, different initial parameters can lead to significantly different optimization paths, contributing to greater performance variance.
>
> 2. As shown in Experiment II (Fig. 2), PALQO generally requires significantly fewer optimization steps than vanilla VQE. For example, to achieve the same target accuracy of $\Delta \leq 0.1$, vanilla VQE may take up to 120 steps, while PALQO—depending on initialization—often converges in just 3 to 4 steps. This results in speedup ratio of 40 and 30, respectively, which largely accounts for the high variance observed.
>
> 3. Additionally, as shown in Figure 3 of the original manuscript, the case with $J/h=2$ is more challenging because it has a smaller energy gap between the ground and first excited states compared to $J/h=0.5, J/h=1$. As a result, PALQO requires more optimization steps to remain competitive, which leads to a lower variance in the speedup ratio.
> Below, we present the iteration counts and variance of the speedup ratio for $J/h=0.5,1$ and $J/h=2$ in the following table.
>
> |                                  | J/h=0.5,1    | J/h=2       |
> |-----------------------|--------------|-------------|
> | Vanilla VQE steps(Ip) | 120,115,130  | 150,140,135 |
> | PALQO steps (Iv)      | 4,3,4        | 26,25,20    |
> | speedup ratio (Ip/Iv) | 40,38.3,32.5 | 5.7,5.6,6.7 |
> | variance  of speedup ratio            | 13.4         | 0.37        |
>
> **L1: The paper includes one sentence about the limitations of this work in the conclusion, which discusses a potential future research direction of this work could be the reduce the high variance in speedup ratios of PALQO as the number of qubits and layers is varied. Other limitations discussed above are not mentioned in the work.**
>
> **Response:** We appreciate the reviewer's suggestion. In the revised manuscript, we expanded the discussion of our work's limitations beyond the high variance in speedup ratios, noting that achieving high accuracy often requires substantial effort in tuning the PINN.
>
> [1] Tilly J, Chen H, Cao S, et al. The variational quantum eigensolver: a review of methods and best practices[J]. Physics Reports, 2022, 986: 1-128.
>
> [2] Cerezo M, Arrasmith A, Babbush R, et al. Variational quantum algorithms[J]. Nature Reviews Physics, 2021, 3(9): 625-644.
>
> [3] Google AI Quantum and Collaborators*†, Arute F, Arya K, et al. Hartree-Fock on a superconducting qubit quantum computer[J]. Science, 2020, 369(6507): 1084-1089.
>
> [4] Kim Y, Eddins A, Anand S, et al. Evidence for the utility of quantum computing before fault tolerance[J]. Nature, 2023, 618(7965): 500-505.
>
> [5] Leimkuhler O, Whaley K B. Exponential quantum speedups in quantum chemistry with linear depth[J]. arXiv preprint arXiv:2503.21041, 2025.
>
> [6] Zhou L, Wang S T, Choi S, et al. Quantum approximate optimization algorithm: Performance, mechanism, and implementation on near-term devices[J]. Physical Review X, 2020, 10(2): 021067.
>
> [7] Shaydulin R, Li C, Chakrabarti S, et al. Evidence of scaling advantage for the quantum approximate optimization algorithm on a classically intractable problem[J]. Science Advances, 2024, 10(22): eadm6761.

---

> ### Author Response · Authors · 2025-08-03
>
> Dear Reviewer 2QJF,
>
> We hope this message finds you well. Thanks for your detailed and constructive feedback which has been invaluable in shaping our work, and we deeply value your insights. We would like to kindly ask whether our latest responses adequately addressed your concerns and clarified the points raised in your previous question. If there are any remaining concerns or areas where further clarification might help, we would be more than happy to address them in the spirit of collaboration and continuous improvement.
>
> Best regards, Authors

---

> > ### Comment · Reviewer_2QJF · 2025-08-04
> >
> > Dear Authors,
> >
> > Thank you for the rebuttal clarifications. By retaining the quantum advantage of VQA, I meant that PALQO itself requires a large classical model to be trained along with the VQA. In that case, why not just train a classical model to optimize the loss function entirely classically? The point of using a quantum model is to not train a classical one. In the case of PALQO, both quantum and classical models have to be trained, thus increasing the amount of work and decreasing any quantum utility for the quantum model. Do you have any further light to shed on this?
> >
> > Thanks.

---

> > > ### Author Response · Authors · 2025-08-05
> > >
> > > **[Q]: Why not just train a classical model to optimize the loss function entirely classically? In the case of PALQO, both quantum and classical models have to be trained, thus increasing the amount of work and decreasing any quantum utility for the quantum model.**
> > >
> > >
> > > **Response:** Thank you for the insightful comment. We agree that developing classical simulators or learning-based surrogates to emulate the expectation values of VQAs is an important research direction [1-4]. However, we emphasize that **this line of work is fundamentally different from ours, as the two approaches have distinct capabilities and objectives**. For clarity, in the remainder of this reply, we first recap the objectives and capabilities of fully classical approaches, and then explain how PALQO advances beyond these methods. All explanations have been updated in the revised manuscript.
> > >
> > > **Fully classical approaches**. Existing literature on fully classical methods for emulating the expectation values of VQAs can be broadly divided into two categories: classical simulators and learning-based surrogates. **Both aim to efficiently approximate the expectation value of a specified ansatz with varying parameters**. However, theoretical results show that these methods cannot efficiently and accurately handle arbitrary ansatzes, highlighting their inherent capabilities and limitations.
> > >
> > > **PALQO**. Let us separately elucidate that PALQO has a distinct objective from fully classical approaches and can efficiently address certain problems that are beyond the reach of such methods.
> > >
> > > $\bullet$ *Objective of PALQO* The objective of PALQO is to predict the optimization trajectory of trainable parameters for a given ansatz, initialization, and problem Hamiltonian. **This task-specific nature distinguishes it from fully classical approaches** and enables it to address certain problems beyond their reach.
> > >
> > > $\bullet$ *Separation from fully classical approaches*. In the following, we provide a concrete example to show the difference between PALQO and fully classical approaches.
> > >
> > > To provide some intuition, we refer to Ref. [5], where the authors demonstrate that their proposed Linear Clifford Encoder (LCE) enables constant-gradient scaling within near-Clifford regions, thereby avoiding barren plateaus in areas where **no entirely classical solutions are known to exist**. This rules out the possibility of efficient fully classical approaches for such cases. By contrast, PALQO can substantially reduce the quantum resources required to optimize these models, making it feasible to experimentally explore their scalability and practical advantages, and enabling large-scale experimental validation of these methods.
> > >
> > >
> > > [1] Pan F, Gu H, Kuang L, et al. Efficient quantum circuit simulation by tensor network methods on modern gpus[J]. ACM Transactions on Quantum Computing, 2024, 5(4): 1-26.
> > >
> > > [2] Fontana E, Rudolph M S, Duncan R, et al. Classical simulations of noisy variational quantum circuits[J]. npj Quantum Information, 2025, 11(1): 84.
> > >
> > > [3] Begušić T, Gray J, Chan G K L. Fast and converged classical simulations of evidence for the utility of quantum computing before fault tolerance[J]. Science Advances, 2024, 10(3): eadk4321.
> > >
> > > [4] Schreiber F J, Eisert J, Meyer J J. Classical surrogates for quantum learning models[J]. Physical Review Letters, 2023, 131(10): 100803.
> > >
> > > [5] Meyer S, Scala F, Tacchino F, et al. Trainability of Quantum Models Beyond Known Classical Simulability[J]. arXiv preprint arXiv:2507.06344, 2025.

---

> > > > ### Comment · Reviewer_2QJF · 2025-08-05
> > > >
> > > > I'm not quite sure why LCE is brought up at this stage. **The PALQO work is entirely evaluated on VQEs**, and as you correctly admitted in the original rebuttal, the **VQE algorithm "currently lacks provable advantages on practical tasks."** As no provable quantum advantage exists for VQE at this stage, and for that matter, even for QAOA, my point is that classical techniques already outperform quantum ones. Thus, if PALQO were a quantum-based technique that accelerates quantum VQE and QAOA, that would be highly encouraging. However, PALQO uses a classical technique to accelerate a quantum implementation, when classical techniques are already known to outperform quantum ones. **So, of course, a classical technique would accelerate a quantum one because it is already faster than the quantum one.** Thus, it is difficult to separate where this acceleration is coming from; my impression is that **it is coming almost entirely from the classical implementation.**
> > > >
> > > > Nonetheless, **I'm not certain what to do with the LCE example.** Is the suggestion to replace all the VQE results in the paper with LCE results because it is shown to support a quantum advantage as opposed to VQE? I'm not sure if that is possible at this point. Of course, one can evaluate any technique designed for VQAs on a VQE, but when the idea is to demonstrate quantum acceleration using a classical bootstrap, my point is simply to choose an algorithm that has a known quantum advantage. **That would help alleviate many of these concerns.**

---

> > > > > ### Author Response · Authors · 2025-08-07
> > > > >
> > > > > **Q1 The PALQO work is entirely evaluated on VQEs, and as you correctly admitted in the original rebuttal, the VQE algorithm "currently lacks provable advantages on practical tasks." As no provable quantum advantage exists for VQE at this stage, and for that matter, even for QAOA, my point is that classical techniques already outperform quantum ones.**
> > > > >
> > > > > **Response:**
> > > > > We thank the reviewer for the valuable comments and the opportunity to clarify our previous rebuttal. We apologize for the imprecise wording that may have conveyed the incorrect impression that the VQE algorithm lacks any potential for quantum advantage. We would like to emphasize that this is not the case.
> > > > >
> > > > > First, the absence of formal theoretical guarantees does not imply the lack of practical utility. VQE is designed as a hybrid quantum-classical approach in which a quantum processor is responsible for preparing parameterized quantum states, while a classical optimizer updates parameters to minimize the expected energy. This division of labor offers a promising paradigm for solving complex quantum many-body problems, particularly on noisy intermediate-scale quantum (NISQ) devices. Even though VQE does not currently offer a proven speedup over classical algorithms across all instances, it may still demonstrate practical benefits in specific domains.
> > > > >
> > > > > Second, VQE targets the ground-state energy estimation of quantum systems, a central challenge in quantum chemistry, condensed matter physics, and materials science. This task is known to be classically intractable in the general case due to the exponential growth of the Hilbert space. Classical approaches, such as full configuration interaction or coupled-cluster methods, face resource bottlenecks as system sizes grow. By contrast, VQE sidesteps this limitation by leveraging quantum hardware to directly prepare and measure quantum states, making it a natural and scalable method for this class of problems.
> > > > >
> > > > > Third, the current lack of demonstrable quantum advantage in VQE is primarily due to hardware limitations rather than deficiencies in the algorithm itself. Today’s quantum devices are constrained by limited qubit counts, short coherence times, and restricted gate fidelities. Many algorithmic innovations designed to achieve quantum advantage cannot yet be fully realized on such devices. This situation parallels early-stage development in other computational paradigms. Notably, recent hardware milestones, including the implementation of scalable quantum error correction [1], the realization of logical qubits [2], and quantum advantage in optimization tasks using Rydberg atom arrays [3], highlight the steady and tangible progress toward practical quantum computation. As hardware improves, algorithms such as VQE are expected to demonstrate their full potential.
> > > > >
> > > > > Fourth, this trajectory is reminiscent of the early development of deep learning, which initially succeeded in practice despite a limited theoretical understanding. It was only after years of sustained hardware innovation (e.g., GPUs) and empirical tuning that deep learning achieved its current dominance. VQE may follow a similar path: it provides a powerful and flexible algorithmic framework whose full capabilities will emerge alongside advances in quantum hardware.
> > > > >
> > > > > In conclusion, although VQE does not yet exhibit a quantum advantage with current limited quantum technology, it remains a leading candidate for achieving one as hardware matures. Enhancing the efficiency and scalability of VQE through classical techniques, such as PALQO, is a worthwhile and timely pursuit. The essence of quantum advantage ultimately lies in quantum computation, but leveraging classical tools to improve convergence, robustness, and practical usability is essential for bridging current limitations and accelerating the realization of quantum advantage in real-world applications.
> > > > >
> > > > > [1] Google Quantum AI. Suppressing quantum errors by scaling a surface code logical qubit. Nature, 614(7949), 676–681.
> > > > >
> > > > > [2] Bluvstein, D., et al. (2024). Logical quantum processor based on reconfigurable atom arrays. Nature, 626, 58–65.
> > > > >
> > > > > [3] Ebadi, S., et al. (2022). Quantum optimization of maximum independent set using Rydberg atom arrays. Science, 376(6598), 1209–1215.

---

> > > > > > ### Author Response · Authors · 2025-08-07
> > > > > >
> > > > > > **Q2 So, of course, a classical technique would accelerate a quantum one because it is already faster than the quantum one. Thus, it is difficult to separate where this acceleration is coming from; my impression is that it is coming almost entirely from the classical implementation.**
> > > > > >
> > > > > > **Response:**
> > > > > > We thank for the reviewer comments.
> > > > > >
> > > > > > Achieving universal quantum computation remains a significant challenge. VQE was originally designed to harness the strengths of classical computing to support quantum devices, reducing the demands on quantum hardware and enabling the realization of quantum advantage even with limited-capability systems. As such, while the core benefits of VQE arise from quantum computation, classical assistance continues to play a crucial role in addressing practical problems both at present and in the near future.
> > > > > >
> > > > > > The main challenge in the optimization process of VQE is the combination of measurement noise from quantum hardware and the inherently complex, non-convex optimization landscape riddled with local minima.
> > > > > >
> > > > > > $\bullet$ **measurement** On any real quantum computer, you can not determine the energy of quantum system with a single, perfect measurement. Instead, you must prepare and measure the state thousands or millions of times (called shots) and compute a statistical average. Due to the inability to apply backpropagation directly on quantum circuits and gradient estimation requires a prohibitively large number of shots, it makes the optimization process incredibly slow and expensive, leading to efficiency issues as system size increases.
> > > > > >
> > > > > > $\bullet$ **local minima** The VQE landscape is a high-dimensional, rugged terrain filled with numerous valleys. The choice of the quantum circuit, or ansatz, heavily influences how rugged this landscape is and how many local minima exist. A classical optimizer, especially a simple one, can easily get trapped in one of these valleys in large-scale cases.
> > > > > >
> > > > > > To address the above issues, we developed PALQO which does **not to improve the computational complexity of VQAs, but to push the limit for large-scale experimentation on current NISQ devices**, thereby facilitating further exploration of the potential advantages of VQA-type algorithms.
> > > > > > It is also a classical-quantum hybrid algortihm, that utilizes quantum device to collect gradient information as training samples and train a PINN-based model based on these data for approximating the optimization dynamics of VQAs to alleviate the quantum measurement burden. Besides, we noticed that PALQO can control the perterbations in optimization trajectory prediction, which sometimes helps escape the local minima.

---

> ### Author Response · Authors · 2025-08-07
>
> **Q3: I'm not certain what to do with the LCE example.**
>
> **Response:**
> The reviewer mentioned that why we not optimize the loss function entirely classically, our answer is we hope but we can not.
>
> The LCE is the example tell that even avoiding barren plateaus (might inherently
> lead to classical simulability, thus limiting the opportunities for quantum advantage which often hinders large-scale VQA), there is still quantum advantage.
>
> The LCE is a classical technique that can modify the first order Clifford structure of Parameterized Quantum Circuits (PQCs), which ensures constant gradient scaling. However, in such loss landscape areas, they proved that there is **no classical simulation technique is known to exist that can efficiently surrogate this landscape region**.
>
> Therefore, in such cases, no classical method can effectively optimize the loss function, making PALQO as a strong candidate. Unlike classical simulators that attempt to emulate VQAs directly, PALQO relies on quantum hardware to estimate gradients and operates in the parameter space rather than the full quantum Hilbert space for training and prediction. Besides, PALQO can substantially reduce the quantum resources required to optimize these models, making it feasible to experimentally explore their scalability and practical advantages, and enabling large-scale experimental validation of these methods.

---

> > ### Author Response · Authors · 2025-08-07
> >
> > Dear Reviewer 2QJF,
> >
> > I hope this message finds you well. As the rebuttal deadline is approaching, we would greatly appreciate your valuable feedback. If there are any remaining concerns or areas where further clarification might help, we would be more than happy to address them in the spirit of collaboration and continuous improvement.
> >
> > We hope that our detailed response and clarifications provided in this rebuttal resolve your concerns, and respectfully believe that some of the critiques may stem from misunderstandings, which we hope we have now clarified.
> >
> > Best, Authors

---

> > > ### Comment · Reviewer_2QJF · 2025-08-07
> > >
> > > Dear Authors,
> > >
> > > Thank you for the explanations. I am aware of the hypothesized use cases and applications of VQE and the motivation of PALQO. I’m not sure if the explanations really answer my questions. I had asked (1) “ it is difficult to separate where this acceleration is coming from; my impression is that it is coming almost entirely from the classical implementation.” Can you provide some analytical/theoretical evidence for VQE retaining its hypothesized advantage with the deployment of PALQO as a classical add-on, other than esoteric explanations? (2) “ Is the suggestion to replace all the VQE results in the paper with LCE results because it is shown to support a quantum advantage as opposed to VQE?” Unless we have results with PALQO evaluated on LCE, I think it’s a moot point to discuss it. Please let me know your answers to these specific questions. They can be very brief and concise.

---

> > > > ### Author Response · Authors · 2025-08-08
> > > >
> > > > **Q1: it is difficult to separate where this acceleration is coming from; my impression is that it is coming almost entirely from the classical implementation.**
> > > >
> > > > **Response:**
> > > >
> > > > We conjecture the reviewer may have meant the following. In the classically intractable cases, each on-device gradient evaluation yields a quantum advantage. However, because PALQO learns the optimization path classically, that portion of the advantage is reduced, i.e. many of those gradient evaluations are no longer performed on the quantum device, and the per-step quantum advantage associated with those evaluations is lost.
> > > >
> > > > More formally, suppose vanilla VQE requires $T$ quantum gradient estimations to reach the target, while PALQO—by learning and predicting the trajectory in parameter space—reduces the number of required quantum gradient estimations to $T'$, with $T > T'$. In this scenario PALQO only exposes VQE’s quantum advantage during the $T'$ quantum evaluations used to collect training samples; the remaining $T-T'$ steps of potential per-step advantage are obviated by the classical surrogate.
> > > >
> > > > Even though the introduction of PALQO can diminish the quantum advantage of VQE (reducing it from $𝑇$-step to $T'$-step advantage), it does not necessarily eliminate it entirely.
> > > >
> > > > On the other hand, while PALQO may attenuate the quantum advantage of vanilla VQE, it plays a crucial role in reducing quantum resource consumption. As the system size grows, the measurement cost becomes prohibitive. For example, in $Fe_2S_2$, energy estimation requires on the order of $O(10^{19})$ shots. Ref [1] estimates that for a 52-qubit $Cr_2$ system, a single energy evaluation would take approximately 25 days, with the full optimization requiring even longer. PALQO can therefore substantially mitigate this measurement overhead.
> > > >
> > > > [1] Tilly J, Chen H, Cao S, et al. The variational quantum eigensolver: a review of methods and best practices[J]. Physics Reports, 2022, 986: 1-128.
> > > >
> > > > **Q2:Is the suggestion to replace all the VQE results in the paper with LCE results because it is shown to support a quantum advantage as opposed to VQE?**
> > > >
> > > > **Response:**
> > > >
> > > > We have added the additional numerical experiments with LCE. The result demonstrate the PALQO is compatible with LCE and still have outperformance in reducing the measurement overhead.
> > > >
> > > > The result is shown in the following table.
> > > >
> > > > | Number of qubits          | 12  |
> > > > |---------------------------|-----|
> > > > | Vanilla VQE steps(Ip)     | 295 |
> > > > | PALQO steps (Iv)          | 14  |
> > > > | speedup ratio (Ip/Iv)     | 21  |
> > > >
> > > > As shown, PALQO requires only 14 times gradient estimation on a quantum device, in contrast to vanilla VQE, which requires 295 such steps to reach ground state, and we claim that PALQO can achieve $295/14\sim 21$ speedup in measurement.

---

> ### Comment · Reviewer_2QJF · 2025-08-08
>
> Dear Authors,
>
> Thank you for the crisp and clear answers.
>
> (1) *"Even though the introduction of PALQO can diminish the quantum advantage of VQE (reducing it from $T$-step to $T'$-step advantage), it does not necessarily eliminate it entirely."* **I agree with this statement completely. Thank you for the acknowledgement.** It would be useful to include a brief discussion on this in the paper to clarify for the reader.
>
> (2) Thank you for also providing some LCE results. They look good.
>
> I have revised my score positively based on our discussion.

---

> > ### Author Response · Authors · 2025-08-08
> >
> > Dear Reviewer 2QJF,
> >
> > We are very grateful to you for your positive evaluation and constructive suggestions. We will add a brief discussion about our discussion in the revised paper. Thank you again!
> >
> > Best regards, Authors

---

### Official Review · Reviewer_xXH5 · 2025-07-02

**Clarity:** 3
**Significance:** 2
**Originality:** 3
**Rating:** 3
**Confidence:** 4

**Summary:**

To reduce the quantum computational resources required for training Variational Quantum Algorithms (VQAs), this paper proposes an approach that leverages a neural network model inspired by Physics-Informed Neural Networks (PINNs) to predict VQA training dynamics. The authors also derive theoretical generalization error bounds for the proposed dynamics prediction model. By applying this method to several quantum-mechanical tasks, the paper demonstrates its practical effectiveness in significantly reducing the necessary quantum computational resources.

**Questions:**

To recommend acceptance, several of the following major questions (though not necessarily all) need to be resolved.
- [Q1] Does this approach help mitigate the Barren Plateau problem?

  For researchers in quantum computing, the primary concern regarding the difficulty of training variational algorithms is the barren plateau problem [1], in which parameter gradients vanish exponentially with respect to the number of qubits. Please clarify whether this study is influenced by barren plateaus, and whether it could potentially contribute to resolving this issue.

- [Q2] How is the reduction in shot count theoretically justified?

  According to Corollary 3.1, training a PALQO model to achieve a given error requires a polynomial-order number of samples with respect to the parameter size. On the other hand, page 3, line 111 states that training a vanilla VQA requires $O(pN_H/\epsilon^2)$ measurements, which scales linearly with $p$. A direct comparison of these two results does not clearly indicate a significant improvement. Please provide a detailed discussion clarifying the extent of the improvement over vanilla VQA that can be inferred from Corollary 3.1.

- [Q3] Can PALCO accurately predict learning dynamics?

  Although sufficient experiments have been conducted regarding improvements in training speed in this study, there is a lack of experiments evaluating how accurately PALCO predicts the learning dynamics of VQA. Comparing PALCO's predicted results to actual learning dynamics, or comparing parameters obtained from TFIM or HQ experiments in this study with optimized parameters from vanilla VQA, would help determine whether predicting learning dynamics via PINN plays an essential role in VQA research.

- [Q4] What contributes to the superiority of PALQO over the other methods?

    In the benchmark comparisons, the LSTM-based method and QuACK were evaluated. However, the discussion is insufficient regarding what factors contribute to PALQO's superiority.

- [Q5] Is this method applicable only to quantum optimization?

  Describing learning dynamics using differential equations and constructing a predictive model via a PINN-like approach seems applicable to classical machine learning contexts as well. Please clarify which specific parts of this method necessitate focusing exclusively on quantum optimization.

[1] McClean, Jarrod R., et al. "Barren plateaus in quantum neural network training landscapes." Nature communications 9.1 (2018): 4812.

**Ethical Concerns:**

["NO or VERY MINOR ethics concerns only"]

**Final Justification:**

The authors have provided very detailed responses, and many of my concerns have been addressed. However, it should be noted that within the quantum computing community, there remains a general skepticism regarding the potential for quantum advantage in variational quantum algorithms. While this manuscript claims to offer a solution by reducing the required number of measurements, I have the following concerns:
- The inherent difficulty of training variational quantum circuits remains unresolved.
- The claimed reduction in measurement counts is not supported by strong theoretical guarantees but rather demonstrated empirically.
- The experiments presented involve training PINNs with only a few dozen samples, leading to high uncertainty in the reported results.

For these reasons, I will maintain my current evaluation.

**Limitations:**

Yes

**Quality:**

2

**Strengths And Weaknesses:**

**Strength**
- The paper proposes a novel approach that experimentally demonstrates significant improvement in addressing one of the critical challenges in training VQAs, specifically the increasing requirement for quantum computational shots.
- Extensive experiments from multiple perspectives have been thoroughly conducted, providing detailed verification of improvements in training speed.
- The manuscript is well-written and thoughtfully structured, making it accessible even to machine learning researchers who may not be familiar with quantum computing.

**Weakness**
- The paper lacks sufficient qualitative discussion about the proposed method. For instance, it does not adequately address how effectively the method tackles significant challenges in VQA, such as Barren Plateau (Q1).
- It remains unclear how PALQO differs from existing methods and what specifically contributes to its advantages (Q2, Q4).
- The validity of the proposed approach is unclear (Q3, Q5). Predicting learning dynamics in VQA appears to be as complex as performing the actual VQA training itself.

---

> ### Author Rebuttal · Authors · 2025-07-29
>
> We sincerely appreciate the reviewer xXH5 for his/her constructive comments, which are invaluable to improve the quality of our submission. For clarity and to better align with the statement in **Weaknesses**, we have reordered the questions [**Q1-Q5**] accordingly. We hope our detailed responses provide helpful clarification and support the reviewer’s continued evaluation of our work.
>
> **[W1 & Q1] The paper lacks sufficient qualitative discussion of PALQO, i.e., how it addresses key challenges in VQAs, such as barren plateau (BP).**
>
> **Response**:  Thanks for the comments. In the remainder of this response, we follow the reviewer’s suggestion to discuss the key challenges of VQAs and the connection between BP and PALQO. All explanations have been incorporated into Appendix B of the revised version.
>
> $\bullet~\underline{\text{Key challenges of VQAs}}$. Optimization efficiency remains the central obstacle to scaling VQAs. On one hand, increasing the number of qubits and circuit depth can lead to the emergence of **BP**, making gradient-based optimization increasingly difficult. On the other hand, due to the no-cloning theorem (Lines 27-28), VQAs must update parameters sequentially to minimize a predefined cost function, which limits the ability to parallelize training. In both cases, the number of measurements required during optimization becomes **prohibitively expensive** for large-scale VQAs.
>
> To address this, a variety of approaches have been developed in recent years to improve the optimization efficiency of VQAs at scale. In tackling BP problems, common strategies include ansatz design [1] and informed initialization [2]. To reduce measurement overhead, typical methods, as reviewed in Appendix B, include measurement grouping, warm-start techniques, and learning-based optimization.
>
> This work primarily focuses on the **latter category**. For this reason, the original version does not elaborate extensively on BP issues, we have append BP review in the updated version. Notably, these approaches are **often complementary and can be integrated to improve overall performance**.
>
> $\bullet~\underline{\text{Relation between BP and PALQO}}$.  PALQO is **complementary to approaches aimed at mitigating barren plateaus**, as both seek to improve the optimization efficiency of VQAs at scale.
>
> To be concrete, consider an example where a BP-free ansatz is applied to estimate the ground state energy of a large molecule such as $Fe_2S_2$. In such cases, the VQE cost function typically decomposes into over 22300 Pauli terms that must be measured individually. To obtain an accurate energy estimate, each of these terms must be sampled 10³ to10⁴ shots, to suppress quantum statistical noise. Even in the absence of BP, assume optimization runs for 1000 steps, the total number of required measurements becomes prohibitively large ($22300×10^3×1000=2.2×10^{10}$). In this context, PALQO plays a crucial role in significantly reducing the measurement cost and shortening the overall runtime, making the optimization process more tractable for large-scale systems.
>
> **[W2 & Q2] It remains unclear how PALQO differs from existing methods. How is the reduction in shot count theoretically justified?**
>
> **Response**:  Let us address the reviewer's concern in sequence.
>
> As for the concern "*How PALQO differs from existing methods*", we follow established conventions in the field [3,4,8,9] by conducting *systematic numerical simulations* across varying problem sizes and types, comparing the performance of PALQO against several baseline models. Notably, unlike prior studies, our work demonstrates heuristic results for systems with **up to 40 qubits**, marking, to our knowledge, the **first such demonstration at this scale**.
>
> As for the concern "*How is the reduction in shot count theoretically justified?*", we address this from two perspectives.
>
> $\bullet$ Recall that it remains a long-standing challenge to provide strong theoretical guarantees for deep learning–enhanced VQEs, as most prior works [3,4,5,6] offer primarily heuristic results. While Corollary 3.1 in the submission does not fully resolve this issue, it represents a **meaningful first step** toward theory-driven algorithm design in this emerging direction.
>
> $\bullet$ **Corollary 3.1 and the measurement cost $O(pN_H/\epsilon^2)$ characterize different aspects of the algorithm and are not directly comparable**. In particular, the results presented in Corollary 3.1 offer theoretical insight into how the number of training examples influences the generalization error of PALQO. As is common in deep learning theory, such bounds are often loose, and the amount of training data required in practice tends to be significantly smaller [7]. As illustrated in Figure 6, PALQO achieves competitive performance even with a limited number of training samples, which is the $\tau$ time-step trajectory data where we set $\tau$ equal to $2$ and $4$, demonstrating its advantage over vanilla VQAs in the measurement cost.
>
> In contrast, the cost $O(pN_H/\epsilon^2)$ refers to the measurement overhead incurred during each iteration of VQE optimization and is independent of the use of PALQO. Typically, the number of the Pauli terms $N_H$ is much larger than $p$, which makes the total measurement cost more strongly influenced by $N_H$ rather than $p$ in quantum chemistry. Thus, PALQO is built for reducing such tremendous measurement costs.
>
> **[W2  & Q4] The paper lacks sufficient discussion on the key factors driving PALQO's superior performance.**
>
> **Response**: PALQO advances baseline models by leveraging PINN to emulate the optimization dynamics of VQEs, incorporating partial derivative information as constraints during training. In the following, we illustrate the advantages of PALQO by integrating results from the original submission with additional experiments conducted in response to the review.
>
> We further evaluate PALOQ with variants where specific loss terms are removed, and compare its performance with LSTM and QUACK, as follows:
>
> | TFIM-20qubits | PALQO | No $L_{p1},L_{p2}$ | No $L_{D}$ | LSTM | QUACK |
> |---------------|-------|-------------------|------------------|------|-------|
> | Speedup Ratio | 30±12 | 2.2±1             | 24±8             | 6±4  | 6±2   |
>
> As can be seen, when constraints $L_{p1}$ and $L_{p2}$ are removed, PALQO degrades into a basic model that merely includes time series information in its input. Its performance is inferior to specialized time series models such as LSTM, and clearly lags behind models like QUACK, which are designed for fitting sequences. These outcomes underscore the critical importance of the PDE residual term as a loss function within the model.
>
> **[W3 & Q3] The validity of the proposed approach is unclear. Can PALCO accurately predict learning dynamics?**
>
> **Response**: To address the reviewer's concern, we have conducted the following experiments to illustrate that PALCO can **indeed accurately predict learning dynamics**.
>
> We conducted simulations on H2, BeH2, and TFIM models. The parameter initialization of the PALQO algorithm is consistent with the main text. In particular, we sampled ten sets of initial parameters for both the VQE and PALQO. After optimization, we compared the deviation of the parameters obtained by VQE and PALQO by calculating their Euclidean norm as shown in the following Table.
>
> |      | H2        | BeH2    | TFIM-8qubits | TFIM-12qubits |
> |------|-----------|---------|--------------|---------------|
> | Norm | 0.03±0.01 | 1.2±0.6 | 0.14±0.03    | 0.192±0.01    |
>
> These results demonstrate that PALQO can accurately predict learning dynamics, achieving a low average deviation. Except for BeH2, PALQO shows very small parameter deviations across models, strongly confirming its accuracy and reliability in predicting optimized parameters.
>
> **[W3 & Q5]  Is this method applicable only to quantum optimization?**
>
> **Response**: Thanks for the comment. While using PINNs to learn training dynamics can be extended to deep learning models, we would like to emphasize that **PALCO is particularly well-suited for VQE optimization tasks**. The reasons are as follows:
> 1) In deep learning, gradients can be efficiently computed via backpropagation. By contrast, the optimization of large-scale VQAs is very challenging, as explained in the reply of [**W1 & Q1**]. As a result, PALQO can provide substantial advancement to improve optimization efficiency.
> 2) The number of parameters $p$ in VQAs is typically much smaller than that in deep neural networks. In PALQO, the width of the PINN is set to $O(cp)$, where $c$ grows linearly with the number of qubits. Accordingly, PALQO remains reasonably sized, and the training complexity is significantly lower than that of applying the same framework to conventional deep learning models.
>
> [1] Park C Y, et al. Hamiltonian variational ansatz without barren plateaus[J]. Quantum, 2024
>
> [2] Zhang K, et al. Escaping from the barren plateau via gaussian initializations in deep variational quantum circuits[J]. NeurIPS, 2022
>
> [3] Di L, et al. Quack: accelerating gradient-based quantum optimization with koopman operator learning. NeurIPS, 36, 2024.
>
> [4] Guillaume V, et al. Learning to learn with quantum neural networks via classical neural networks. arXiv preprint arXiv:1907.05415
>
> [5] Cervera-Lierta A, et al. Meta-variational quantum eigensolver: Learning energy profiles of parameterized hamiltonians for quantum simulation[J]. PRX Quantum, 2021
>
> [6] Karim A, et al. Fast and Noise-aware Machine Learning Variational Quantum Eigensolver Optimiser[J]. arXiv:2503.20210.
>
> [7] Jiang Y, et al. Fantastic generalization measures and where to find them[J]. arXiv:1912.02178, 2019.
>
> [8] Meng F, et al. Conditional Diffusion-based Parameter Generation for Quantum Approximate Optimization Algorithm[J]. arXiv:2407.12242
>
> [9] Khairy S et al. Learning to optimize variational quantum circuits to solve combinatorial problems[C]//AAAI. 2020

---

> > ### Comment · Reviewer_xXH5 · 2025-08-03
> >
> > Thank you for your rebuttal and clarifications.
> >
> > **[W2 & Q4] [W3 & Q3] [W3 & Q5]**
> > \
> > Thank you for your kind response. It makes sense to me.
> >
> > **[W1 & Q1]**
> > \
> > I understand that this study focuses on reducing the number of measurements, which complements efforts to address the barren plateau problem. However, given that training difficulties and scalability are intrinsic challenges in VQE, a significant limitation of this work is that it does not contribute to resolving these fundamental issues.
> >
> > **[W2 & Q2]**
> > \
> > Corollary 3.1 states that we need $O(p)$ (or is it $O(p N_H)$?) training examples to achieve an error tolerance $\gamma$. Therefore, collecting these examples requires $O(p) \times (\text{number of measurements per example})$ measurements in total. In other words, to skip certain VQE training steps with PALQO, we must perform $O(p)$ measurements to build the training dataset. By contrast, vanilla VQE needs $O(p N_H)$ measurements to reach the error tolerance $\epsilon$. Consequently, it appears there is no measurement-count speed-up. Am I mistaken? (I realize these bounds are not tight and that your paper shows a measurements reduction empirically.)
> >
> > ---
> > Additionally, I would like to ask the following follow-up questions:
> >
> > **[Q6]**
> > In Figure 6, PALQO appears to achieve better results when trained on fewer data points. Could you explain why this happens?
> >
> > **[Q7]**
> > Is it common practice within the PINN community to train PINNs using only 2 or 4 data points? From a conventional machine-learning perspective, that sample size seems extremely small.

---

> ### Author Response · Authors · 2025-08-04
> **Response for **[W1&Q1]****
>
> **[W1 & Q1]** I understand that this study focuses on reducing the number of measurements, which complements efforts to address the barren plateau problem. However, given that training difficulties and scalability are intrinsic challenges in VQE, a significant limitation of this work is that it does not contribute to resolving these fundamental issues.
>
> **Response**: We appreciate the reviewer’s observation regarding the scope of our contribution in relation to the broader challenges in VQE, particularly the barren plateau (BP) problem. Below, we address this from two complementary perspectives:
>
> **Scopre and Positioning**
>
> - We fully acknowledge that the BP phenomenon represents a fundamental obstacle to the scalability of VQE, and we agree it remains one of the most critical open problems in the field. A recent Nature Reviews Physics article [1], provides a comprehensive survey of existing progress and highlights the importance of exploring both algorithmic opportunities and hardware-aware constraints in VQA design.
>
> - While our work does not directly resolve the core difficulty of BP, we believe it still offers meaningful value. As emphasized in Ref. [1], *reducing the need for large-scale heuristic implementations is itself an important goal for near-term quantum computing*. By significantly lowering the number of measurements required during training, PALQO enables more practical and resource-efficient evaluation of VQE models.
>
> - Furthermore, recent studies have proposed VQA variants that are **provably BP-free and may exhibit quantum advantages**. PALQO reduces the experimental burden needed to validate and scale such promising approaches, thereby expanding the scope of what can be feasibly tested on near-term devices.
>
> **Complementarity to BP-Free Variational Models**
>
> To further clarify the contributions of PALQO, we offer concrete examples showing how it can strengthen the practicality and scalability of recent BP-free approaches:
> - Smart initialization Methods: The recent two papers [2,3] ensure that the variational landscape exhibits **substantial gradients** near a good initialization point. PALQO can dramatically reduce the measurement cost required to exploit this local structure, thereby enabling large-scale experimental validation of these methods.
>
> - Trainability Beyond Classical Simulability: The work [4] introduces a Linear Clifford Encoder (LCE) that guarantees constant-gradient scaling within near-Clifford regions. As in previous examples, PALQO can reduce the quantum resource demands required to optimize such models, **making it feasible to experimentally explore their scalability and practical advantages**.
>
> In summary, while PALQO does not directly solve the barren plateau problem, it addresses a key complementary bottleneck: measurement overhead. As with prior AI conference papers on AI-enhanced VQEs [5-10], while the results may not justify a strong acceptance on their own, they offer novel insights for both the quantum machine learning and variational quantum computing communities. We hope the reviewer will find that **despite some limitations, PALQO offers a meaningful contribution, making it a valuable addition to the conference program**.
>
> [1] Larocca, M., Thanasilp, S., Wang, S. et al. Barren plateaus in variational quantum computing. Nat Rev Phys 7, 174–189 (2025).
>
> [2] Zhang, Kaining, et al. "Escaping from the barren plateau via gaussian initializations in deep variational quantum circuits." Advances in Neural Information Processing Systems 35 (2022): 18612-18627.
>
> [3] Puig, Ricard, et al. "Variational quantum simulation: a case study for understanding warm starts." PRX Quantum 6.1 (2025): 010317.
>
> [4] Meyer, Sabri, et al. "Trainability of Quantum Models Beyond Known Classical Simulability." arXiv preprint arXiv:2507.06344 (2025).
>
> [5] Luo, Di, et al. "Quack: Accelerating gradient-based quantum optimization with koopman operator learning." Advances in Neural Information Processing Systems 36 (2023): 25662-25692.
>
> [6] Dai, Zhongxiang, et al. "Quantum bayesian optimization." Advances in Neural Information Processing Systems 36 (2023): 20179-20207.
>
> [7] Nicoli, Kim, et al. "Physics-informed bayesian optimization of variational quantum circuits." Advances in Neural Information Processing Systems 36 (2023): 18341-18376.
>
> [8] Qian, Yang, et al. "MG-Net: Learn to Customize QAOA with Circuit Depth Awareness." Advances in Neural Information Processing Systems 37 (2024): 33691-33725.
>
> [9] Wu, Huanjin, Xinyu Ye, and Junchi Yan. "Qvae-mole: The quantum vae with spherical latent variable learning for 3-d molecule generation." Advances in Neural Information Processing Systems 37 (2024): 22745-22771.
>
> [10] Ostaszewski, Mateusz, et al. "Reinforcement learning for optimization of variational quantum circuit architectures." Advances in neural information processing systems 34 (2021): 18182-18194.

---

> ### Author Response · Authors · 2025-08-04
> **Response for **[W2 & Q2]**(1)**
>
> **[W2 & Q2]**： Whether PALQO offers a true measurement reduction, given that collecting the required $O(p)$ training examples entails $O(p) \times (measurements per example)$, which may be comparable to the O(pN_H) measurements needed by vanilla VQE to achieve the same error tolerance—potentially eliminating any asymptotic advantage.
>
> **Response:** Thanks for the comments. We would like to clarify that PALQO **indeed offers a measurement reduction**. The potential confusion may stem from the fact that the measurement cost for PALQO, **both in the theoretical result of Corollary 3.1 and in its implementation, is expressed over $T$ optimization steps**. In contrast, the measurement cost of vanilla VQE is typically reported **per iteration**. To clarify this distinction, we summarize the measurement reduction offered by PALQO in the table below, where the notation $m$ is the number of the training and data recollection of PINN and $\tau$ refers to the number of samples used in each training session of the PINN. (Recap: in practice, PALQO adopts a sliding-window approach thus it will recollect the data and retrain before finding ground state)
>
> |                      Vanilla VQE                     |                       |   | PALQO                                                    |                               |
> |:----------------------------------------------------:|-----------------------|---|----------------------------------------------------------|-------------------------------|
> | Measurement-counts  for gradient estimation per iter |  $O(pN_H/\epsilon^2)$ |   | Measurement-counts  for gradient estimation per examples |      $O(pN_H/\epsilon^2)$     |
> |               Total number of iteration              |           T           |   |    Total number of the  training and data recollection   |               m               |
> |                                                      |                       |   |   Number of samples per  training and data recollection  |             $\tau$            |
> |               Total measurement-counts               | $O(TpN_H/\epsilon^2)$ |   |                 Total measurement-counts                 | $O(\tau m pN_H/\epsilon^2)$ |
>
> We next elucidate the required number of measurements for each approach.

---

> ### Author Response · Authors · 2025-08-04
> **Response for **[W2 & Q2]**(2)**
>
> **Vanilla VQE**. The reviewer's analysis of the required number of measurements for vanilla VQE is correct. That is, the measurement cost yields $O(TpN_H/\epsilon^2)$, as each optimization step involves estimating gradients using $O(pN_H/\epsilon^2)$ measurements.
>
> **PALQO**. For clarity, let us first recap the goal of PALQO. Given a specified Hamiltonian and ansatz, PALQO aims to predict the optimization dynamics of VQE using physics-informed neural networks (PINNs). **Training the PINN amounts to learning and mimicking the trajectory of parameter updates for the corresponding vanilla VQE**. Once trained, PALQO can predict future optimization steps **entirely on the classical side**, without requiring additional quantum measurements. Consequently, quantum measurements are only needed during the training phase, while inference proceeds fully classically.
>
> Building on this understanding, we next elucidate the meaning of Corollary 3.1 and clarify the measurement requirements for implementing PALQO.
>
> $\underline{\text{Interpretation of Corollary 3.1}}$. Recall that Corollary 3.1 provides a generalization bound of PALQO in terms of the number of training examples $\tau$. More specifically, **$\tau$ corresponds to the optimization trajectory, i.e., the number of VQE iterations from which gradient information is collected via quantum hardware**. Each training example in $\tau$, corresponding to the acquisition of gradient information at a single iteration, requires $O(p N_H / \epsilon^2)$ measurements.
>
> The objective of Corollary 3.1 is to show that the generalization error of PALQO can be bounded. The derived bound depends on the number of parameters $p$, the number of training samples $\tau$, as well as the depth $L$ and width $W$ of the PINN. This corollary is **not directly related to measurement reduction**.
>
> $\underline{\text{Measurement requirements for implementing PALQO}}$. Let us briefly recap the implementation of PALQO. To improve practical performance, PALQO adopts a sliding-window approach, similar to the strategy used in QuACK. Once the predicted optimization trajectory converges, additional data samples are collected from the quantum hardware, and PALQO is retrained using the updated dataset. This procedure is **repeated iteratively $m$ times until the loss function of PALQO converges**.
>
> We now analyze the required number of measurements of PALQO. Suppose the training and data recollection process is repeated $m$ times (typically $m<10$ in practice)  before the PALQO loss function converges. Now for each time, we assume $\tau=O(p)$ training examples are employed to train PALQO, where $O(pN_H/\epsilon^2)$ measurements are adopted per example. Then, the total number of measurements for PALQO is $O(mp^2N_H/\epsilon^2)$.
>
> Consequently, **the reduction in total measurement cost over $T$ steps is approximately $O((T-mp)pN_H/\epsilon^2)$**. In practice, since $T\gg m*p$, PALQO provides a substantial measurement reduction.
>
> [1] Luo, Di, et al. "Quack: Accelerating gradient-based quantum optimization with koopman operator learning." Advances in Neural Information Processing Systems 36 (2023): 25662-25692.

---

> ### Author Response · Authors · 2025-08-04
> **Response for **[Q6 & Q7]****
>
> **[Q6&Q7]** In Figure 6, PALQO appears to achieve better results when trained on fewer data points. Could you explain why this happens? Is it common practice within the PINN community to train PINNs using only 2 or 4 data points? From a conventional machine-learning perspective, that sample size seems extremely small.
>
> **Response:** Thank you for the comments. We address the reviewer’s concerns from two perspectives. First, we clarify the number of training examples used in each setting of Figure 6. Then, we provide intuition for why PALQO is able to perform well with a limited number of training examples.
>
>
> **The number of training examples in Figure 6**. We follow the definition of $\tau$ and the implementation details of PALQO provided in our response to [**W2 & Q2**] to clarify the number of training examples used by PALQO in Figure 6.
>
> Recall that for PALQO, the training and data recollection process is repeated $m$ times (typically $m<10$ in practice) before the PALQO loss function converges. **The lengend in Figure 6 only exhibit the adopted number training examples of at each iteration**. The employed number of iterations and the total number of training examples are summarized in the following table. We appologize for this misleading and have revised the manuscript to explain the total number of training examples in the caption of Figure 6.
>
> | m          | 10 | 5  | 3  |
> |------------|----|----|----|
> | $\tau$     | 2  | 4  | 8  |
> | number of training examples | 20 | 20 | 24 |
>
> **Why is PALQO able to perform well with fewer training examples?**. We emphasize that the number of training examples required at each iteration (repeated $m$ times) depends on the complexity of the optimization trajectory associated with the specific Hamiltonian–ansatz pair, which determines how difficult it is for PALQO to learn the underlying dynamics.
>
> To provide some intuition, we refer to Ref. [1], which shows that with a well-chosen initialization, the local loss landscape can be approximately convex. This structure allows PALQO to accurately learn the optimization trajectory using only a small number of training examples. As further supported by the numerical results appended in our response to [W2 & Q4], the loss constraints $L_{P1}$ and $L_{P2}$ guide PALQO’s predicted trajectory to closely match the ground-truth optimization path.
>
>
> [1] Puig, Ricard, et al. "Variational quantum simulation: a case study for understanding warm starts." PRX Quantum 6.1 (2025): 010317.

---

> > ### Comment · Reviewer_xXH5 · 2025-08-05
> >
> > I would like to thank the authors for their detailed responses, which have significantly clarified the focus and contributions of the current work. However, my evaluation remains unchanged at this stage.
> >
> > My primary concern continues to be scalability. Specifically, the theoretical justification provided for the reduction in the required number of measurements remains insufficiently strong, and the empirical validation presented in the manuscript is limited to relatively small-scale training data. Consequently, it is still unclear how the proposed methodology could be effectively scaled to handle larger and more complex circuits (beyond classical limitations) in practical scenarios.

---

> ### Author Response · Authors · 2025-08-05
>
> **Q8: The primary concern continues to be scalability. Specifically, the theoretical justification provided for the reduction in the required number of measurements remains insufficiently strong, and the empirical validation presented in the manuscript is limited to relatively small-scale training data. Consequently, it is still unclear how the proposed methodology could be effectively scaled to handle larger and more complex circuits (beyond classical limitations) in practical scenarios.**
>
> **Response:** Thank you for the comments.
> 1. We agree the reviewer's comments that the previous theoretical analysis of measurement reduction provided an estimate of the scaling behavior, rather than a tight bound. To obtain a more precise estimation, we need to analyze the number of iterations $T$ required for VQE optimization. However, this remains a challenging unsolved open problem, as $T$ depends on various factors, including the choice of initialization, the optimization algorithm, and the specific ansatz etc. Thus, it is hard to have an accurate measurement reduction estimation in current, unfortunately.
>
> 2. In the original manuscript, we evaluated the scalability of PALQO through experiments on quantum systems ranging from 12 to 36 qubits. Specifically, we assessed the speedup of PALQO compared to vanilla VQE in reaching the ground state. The results, presented in the table below, show that PALQO consistently achieves satisfactory speedup performance.
>
> | System size       | 12 | 20 | 28 | 36 |
> |-------------------|----|----|----|----|
> | m                 | 10 |  12| 13 | 15 |
> | $\tau$            | 2  | 3  | 3  |  3 |
> | Number of samples |  20  |  36  |  39  |  45  |
> | number of variational parameters| 72 | 120 | 168 | 216 |
> | Speedup ratio     | 27 | 29 | 23 | 26 |
>
> From the results shown in the table, it is evident that the number of training samples required by PALQO does not grow exponentially with system size. Moreover, we observe that the introduction of physical constraints allows PALQO to accurately predict the optimization trajectory over the local loss landscape using only a small number of training samples.
>
> 3. For cases that cannot be classically simulated, we would like to clarify that PALQO does not compute the gradient information in VQA through classical simulation. Instead, this gradient information is obtained directly from quantum hardware. As a result, the dimension and sample size of PALQO’s training data are significantly smaller than those required for classically simulating a quantum system that scales exponentially as $2^n$. In contrast, the dimension of the training data depends only on the number of parameters $p$ in the variational quantum circuit. Typically, this number grows linearly or polynomially with the number of qubits, meaning that the size of PALQO’s training data also scales linearly or polynomially with system size.
> As PALQO does not operate in the full quantum state space; rather, it learns a trajectory in the parameter space, allowing it to bypass the exponential complexity associated with simulating the entire quantum system. Therefore, PALQO can effectively scaled to handle larger systems.
>
> We sincerely thank Reviewer xXH5 for your time and for providing constructive feedback that has helped us improve our manuscript. We have carefully considered all comments and provide a point-by-point response. In light of these detailed responses and our planned revisions, we respectfully ask if the reviewer would be willing to reconsider your evaluation of our work.

---

> > ### Author Response · Authors · 2025-08-07
> >
> > Dear Reviewer xXH5,
> >
> > I hope this message finds you well. As the rebuttal deadline is approaching, we would greatly appreciate your valuable feedback. If there are any remaining concerns or areas where further clarification might help, we would be more than happy to address them in the spirit of collaboration and continuous improvement.
> >
> > For the reviewers concern about scalability, we respectfully note that **reducing the measurement burden is a critical step toward improving scalability** in real-world quantum applications. Besides, we believe our work is the first one that addresses training efficiency issue over 30-qubits quantum system and empirically demonstrate the advantages in reducing measurement overhead, which is also valued by reviewer KJmn.
> >
> > We hope that our detailed response and clarifications provided in this rebuttal resolve your concerns, and respectfully believe that some of the critiques may stem from misunderstandings, which we hope we have now clarified.
> >
> > Best, Authors

---

> > > ### Author Response · Authors · 2025-08-08
> > >
> > > Dear Reviewer xXH5,
> > >
> > > I hope this message finds you well. We would greatly appreciate your valuable feedback. As the rebuttal deadline approaches, we are eager to learn if there are any remaining concerns or points requiring further clarification. Within the limited time left for discussion, we would be glad to address them in the spirit of collaboration and continuous improvement.
> > >
> > > We believe that some of the critiques may stem from misunderstandings, which we have sought to clarify. We hope that the detailed responses and explanations provided in our rebuttal sufficiently address your concerns, as they have for other reviewers.
> > >
> > > Best regards, Authors

---

### Official Review · Reviewer_KJmn · 2025-07-03

**Clarity:** 3
**Significance:** 3
**Originality:** 3
**Rating:** 5
**Confidence:** 4

**Summary:**

This paper introduces a novel perspective on the optimization dynamics of variational quantum algorithms (VQAs) by modeling them through partial differential equations (PDEs). It establishes a PDE-based framework that governs VQA parameter updates and enables efficient classical prediction via physics-informed neural networks (PINNs). Beyond the theoretical insight, the authors propose a practical protocol to reduce quantum measurement costs during training by leveraging this classical surrogate model. Importantly, unlike many prior works limited to small-scale simulations, this study validates its approach empirically on systems with up to 40 qubits, offering meaningful insights into the scalability of PDE-based optimization for VQAs. The principal contribution lies in showcasing a promising classical strategy to accelerate large-scale VQA training, addressing one of the critical bottlenecks in near-term quantum computing.

**Questions:**

1. For the quantum chemistry benchmarks, do all cases achieve chemical accuracy (i.e., ~1e-3 kcal/mol)? If not, which systems fall short, and why?
2. While the experiments extend up to 40 qubits, they are restricted to shallow circuits (e.g., 3-layer HEA). Given that such circuits may contain relatively few parameters within the effective light cone, how strong is the evidence for scalability? More discussion on this point would be helpful.
3. In Line 111, the framework is evaluated using standard measurement protocols. Have the authors considered integrating classical shadows or other advanced measurement schemes? I am happy to see more discussions about this direction.
4. In Appendix F.3, while the noisy simulations are appreciated, it remains unclear whether the method maintains its predictive performance across varying noise levels. Evaluations under different noise profiles are necessary.

**Ethical Concerns:**

["NO or VERY MINOR ethics concerns only"]

**Limitations:**

Yes.

**Paper Formatting Concerns:**

No.

**Quality:**

4

**Strengths And Weaknesses:**

Strengths:
- The paper makes a compelling theoretical contribution by connecting the optimization trajectories of VQAs with dynamical systems governed by PDEs. This analogy allows the use of PINNs to predict parameter evolution classically, thereby circumventing the need for repeated quantum measurements—a major constraint in training variational circuits—while sidestepping limitations such as the no-cloning theorem.
- The proposed framework is general and applicable across VQA variants. It directly targets a practical bottleneck—quantum measurement overhead—and provides a protocol that can adapt to existing VQA pipelines.
- The work is backed by extensive numerical studies, including simulations on systems with up to 40 qubits. These empirical results provide strong support for the scalability and practical utility of the approach, which is rare among works in this domain.

Weakness:
- The initialization strategy for the variational parameters, i.e., uniform sampling from the [0,1] range as used in the QuACK benchmark, may limit the generality of the results. If the performance of PALQO (the proposed method) is sensitive to this specific range, it raises concerns about robustness. Additional experiments with varied initialization schemes would strengthen the conclusions.
- In Section 4.2, while the authors quantify the reduction in measurement overhead, the analysis is based on single instances for each system size. It is unclear whether these reductions persist across different Hamiltonians or circuit depths. A broader statistical analysis would be beneficial to evaluate the consistency of the advantage.

---

> ### Author Rebuttal · Authors · 2025-07-29
>
> We sincerely thank Reviewer KJmn for your constructive feedback and valuable suggestions.  We have indexed the points mentioned under **Weaknesses** as **W1** and **W2**, and **Questions** as **Q1** to **Q4**. We hope our responses clarify the contributions and assist the reviewer in reassessing our submission.
>
> **W1: Whether the robustness of PALQO is due to its fixed [0,1] parameter initialization and suggests testing with varied schemes to validate generality.**
>
> **Response:** Thanks for the comment. To address the reviewer's concern, we perform extra experiments on different initialization strategy. Here we change the sampling from (0, 1) to ($-\pi$, $\pi$), and the result is as follows.
>
> |               | TFIM-12 | BeH2  |
> |---------------|------|-------|
> | Speedup Ratio | 18±9 | 23±12 |
>
> The results show that PALQO can still achieve good performance under various initialization schemes.
>
> **W2: The suggestion about broader statistical analysis to verify whether measurement reductions hold across various Hamiltonians and circuit depths.**
>
> **Response:** Thank you for your insightful suggestion to extend the statistical analysis of measurement reduction across different system sizes. We have followed the suggestion to add related experiments as blow.
>
> |       | TFIM-12qubits, depth=3 | TFIM-8qubits, depth=3 | TFIM-12qubits, depth=5 | TFIM-8qubits,depth=5 |
> |-------|------------------------|-----------------------|------------------------|----------------------|
> | VQE   | $3.882 \times 10^8$             | $1.688 \times 10^8$            | $6.649 \times 10^8$             | $2.813 \times10^8$           |
> | PAlQO | $5.432 \times 10^7$             | $2.362 \times 10^7$            | $9.053 \times 10^7$             | $8.016 \times10^7$           |
>
> This result shows that even in systems of different sizes and depths, PALQO can still effectively reduce measurement cost.
>
> **Q1: For the quantum chemistry benchmarks, do all cases achieve chemical accuracy (i.e., ~1e-3 kcal/mol)? If not, which systems fall short, and why?**
>
> Thanks for your comment. We apologize for not presenting the final model accuracy achieved after utilizing PALQO to accelerate optimization in the original manuscript. In all cases, the PALQO can achieve the chemical accuracy.
>
> **Q2: While the experiments extend up to 40 qubits, they are restricted to shallow circuits (e.g., 3-layer HEA). Given that such circuits may contain relatively few parameters within the effective light cone, how strong is the evidence for scalability? More discussion on this point would be helpful.**
>
> **Response:** We appreciate the reviewer's comment and suggestion. Here, we would like to clarify that even quantum circuits with shallow depth, when constructed from Clifford and T gates, can still be classically intractable. This is because such circuits evolve quantum states into regions of Hilbert space that lack the structure necessary for efficient classical simulation. We also followed the suggestion to add such discussion in revised manuscript.
>
> **Q3: In Line 111, the framework is evaluated using standard measurement protocols. Have the authors considered integrating classical shadows or other advanced measurement schemes? I am happy to see more discussions about this direction.**
>
> **Response:** Thanks for the suggestion. We agree that it would be beneficial to integrating the classical shadows or advanced measurement schemes to further enhance the performance. This aligns with our experiment on measurement grouping to reduce the number of distinct measurements, as presented in Section F.4 of the Appendix. We have followed this suggestion and extend the discussion of integrating other advanced measurement schemes to further improve the performance in F.4.
>
> **Q4: In Appendix F.3, while the noisy simulations are appreciated, it remains unclear whether the method maintains its predictive performance across varying noise levels. Evaluations under different noise profiles are necessary.**
>
> **Response:** Thanks for the comment. To address the reviewer's concern about the performance of PALQO under various noise level, we have added related experiments as follows.
> |               | 10% depolarizing noise | 20% depolarizing noise | Amplitude Damping Channel |
> |---------------|------------------------|------------------------|---------------------------|
> | Speedup Ratio | 10±6                   | 6±3                    | 13±5                      |
>
> In the main text, we only considered 5% depolarization noise. Here, we added 10% and 20% depolarization noise, as well as an Amplitude Damping Channel. The final results show that PALQO has strong robustness under different noise models.

---

> > ### Comment · Reviewer_KJmn · 2025-08-05
> >
> > I thank the authors for their detailed rebuttal.
> >
> > After considering the concerns raised by other reviewers, I prefer to maintain my positive score. While this work may not directly address the barren plateau problem, which is widely regarded as one of the most significant challenges in VQA, it offers a novel perspective on improving the scalability and practicality of VQAs. As stated by the authors, the proposed method is complementary to existing BP-mitigation approaches.
> >
> > Overall, I believe this work holds value, particularly given its empirical advantage in reducing measurement overhead. Consistent with prior QML research published at top AI conferences, it presents ideas that are both timely and relevant, especially for QML community.

---

> > > ### Author Response · Authors · 2025-08-07
> > >
> > > Dear Reviewer KJmn,
> > >
> > > We sincerely thanks for your positive evaluation and constructive feedback. Your encouraging remarks regarding the significance and potential impact of our work are greatly appreciated. We are grateful for your time and effort in reviewing our manuscript.
> > >
> > > Best, Authors

---

### Official Review · Reviewer_KUyt · 2025-07-04

**Clarity:** 2
**Significance:** 2
**Originality:** 2
**Rating:** 4
**Confidence:** 3

**Summary:**

The paper proposes a learning-based method to predict the dynamics of parameter evolution in variational quantum circuits. It rewrites the dynamics of variational quantum circuits’ training as dynamics of a partially differentiable equation and employs the physics-informed neural network to predict the dynamics of PDE. This framework is scalable and helps accelerate the training of VQA. An extensive set of experiments demonstrates the efficacy of the proposed framework.

**Questions:**

I wonder if the PALQO will always converge for VQE. Can you provide a theorem guaranteeing the convergence on VQEs of the proposed method?

**Ethical Concerns:**

["NO or VERY MINOR ethics concerns only"]

**Final Justification:**

My three concerns are adequately resolved in the rebuttal.

**Limitations:**

No.

**Paper Formatting Concerns:**

No.

**Quality:**

3

**Strengths And Weaknesses:**

Pros
1. The motivation of PALQO roots in the heavy resources needed in VQA training. PALQO effectively exploits the patterns in the VQA training dynamics to reduce the number of samples needed, which is clever.
2. The experiments are comprehensive and clearly demonstrate the advantages of PALQO.

Cons
1. From my observations of Figure 2., in many cases PALQO has not converged to zero ΔE. What are the causes of this?
2. I feel there is a lack of discussion on the experiment settings. How do you make sure that the comparison of PALQO and other methods is fair? I.e., what metrics are kept the same when you report the required number of measurements for different methods, especially when you cannot obtain ~0 ΔE for each experiment?

---

> ### Author Rebuttal · Authors · 2025-07-29
>
> We sincerely thank Reviewer KUyt for their constructive feedback and valuable suggestions. We have carefully addressed all concerns raised. For clarity, we index the points mentioned under **Weaknesses** as **W1** and **W2**, and **Questions** as **Q1**. We hope our responses clarify the contributions and assist the reviewer in reassessing our submission.
>
> **W1: From my observations of Figure 2, in many cases PALQO has not converged to zero $\Delta E$. What are the causes of this?**
>
> **Response**: Thanks for the comments. To address the reviewer’s concern, we first clarify the purpose of the experiments in Fig. 2 and then provide additional simulation results demonstrating PALQO’s ability to converge to zero $\Delta E$.
>
> *$\bullet$ $\underline{\text{Purpose related to Fig. 2}}$*. The primary goal of Fig. 2 is to show the performance advantage of PALQO. In particular, the evaluation metric is **speedup ratio**, which amounts to quantifying the ratio of the number of iterations required by vanilla VQE to that of the baseline method, rather than the energy difference $\Delta E$ (as the reviewer noted). Under this metric, the results in Fig. 2 indicate that PALQO consistently outperforms vanilla VQE, achieving up to a 30× speedup.
>
> It is noteworthy that **the evaluation of PALQO’s advantage over baseline models is not limited to a single metric**. The experiments in Fig. 2 focus on performance comparison under a limited quantum resource budget (i.e., a fixed number of iterations), which reflects a common practical constraint.
>
> *$\bullet$ $\underline{\text{Ability to converge to zero $\Delta E$}}$*.  To better address the reviewer's concerns, we conducted additional simulations to demonstrate PALQO's ability to achieve zero $\Delta E$. In this setting, the evaluation metric is the energy difference from the exact ground-state energy *after convergence*, with no constraint on the number of iterations.
>
> The experimental settings are as follows. We conducted simulations on a 12-qubit TFIM with HEA and a 14-qubit BeH\(2\) with UCCSD ansatz. The hyperparameter settings for these are consistent with those detailed in the main text.
> |               |            | TFIM                 | BeH2                 |
> |---------------|------------|----------------------|----------------------|
> | Vanilla VQE   | $\Delta E$ | $4.266\times10^{-2}$ | $1.271\times10^{-3}$ |
> |               | iterations($I_p$) | 334                  | 676                  |
> | PALQO         | $\Delta E$ | $4.264\times10^{-2}$ | $1.269\times10^{-3}$ |
> |               | iterations($I_v$) | 30                   | 29                   |
> | Speedup ratio |  $I_p/I_v$       | 11.13                | 23.31                |
>
> As shown in the above Table, once vanilla VQE is capable of converging to a small $\Delta$, then *PALQO can do this either with the extra measurement reduction*.
>
> **W2: The submission lacks the details regarding the experimental settings and fairness of the comparisons. Specifically, when comparing PALQO with other methods, what metrics are kept the same when reporting the required number of measurements for different methods?**
>
> **Response**: We highly agree with the reviewer's viewpoint, such that it is crucial to ensure fair comparisons. To this end, **in the original submission, we adopt different metrics to systematically evaluate the capabilities and limitations of PALQO compared to other baseline models**. For convenience, we summarize the adopted metrics and experimental settings of each figure in the following Table.
>
> | condition    |  in main text | metric            | description                                                                                                    |
> |-------------|--------------|-------------------|-------------------|
> | iteration=20 | Figure 2     | ΔE                | how close the estimated energy $\hat{E}$ is close to target energy $E$, $\Delta E=\|\hat{E}-E\|$               |
> | ΔE~1e-3 | Figure 2     | speedup ratio     | The ratio ($I_p/I_v$) presents the number of iterations required by vanilla VQE($I_p$) relative to that of PALQO($I_v$). |
> | ΔE~1e-3 | Table 1      | measurement count | the number of measurements incurred during the optimization  |
>
> We use the above metrics under specific conditions to evaluate the performance of PALQO and other baseline models separately. The detailed settings are as follows:
> 1. Since both PALQO and the baselines can reach $\Delta E \sim 10^{-3}$, we fix the number of iterations to be 20 and compare the lowest $\Delta E$ each model can achieve within this limit. (left subfigure in each column of Figure 2).
> 2. By fixing $\Delta E \sim 10^{-3}$, we use the speedup ratio as a metric to evaluate PALQO’s improvement in training efficiency relative to the baselines (right subfigure in each column of Figure 2).
> 3. We calculate the total number of measurements throughout the entire optimization process until $\Delta E$ of the model approach $10^{-3}$ (Table 1).
>
> **Q1: I wonder if the PALQO will always converge for VQE. Can you provide a theorem guaranteeing the convergence on VQEs of the proposed method?**
>
> **Response**: We acknowledge the reviewer’s concern regarding the lack of strong theoretical guarantees for deep learning–enhanced VQEs, particularly in ensuring robustness and interpretability. Indeed, this remains a **long-standing challenge** in the field, as most prior works [1][5] offer primarily **heuristic results**. While Corollary 3 in our submission does not fully resolve this issue, we believe our work represents a meaningful **first step** toward theory-driven algorithm design in this emerging direction.
>
> To better address the reviewer's concern, we briefly summarize the main results and provide a proof sketch showing that, in the extreme case, PALQO is guaranteed to converge for VQE. We would be happy to share additional proof details during the discussion session.
>
> Assume that PALQO is applied to learn the optimization dynamics of a VQE model in the overparameterized regime with a sufficiently small learning rate $\eta$. Suppose further that the training dataset is sufficiently large and the employed PINN is also overparameterized. Under these conditions, deep learning theory guarantees that PALQO will exhibit a small expected risk $\epsilon_0$, and is therefore theoretically capable of accurately tracking the training dynamics of the overparameterized VQE.
>
> The sketch of proof is as follows. According to the result of Theorem 1 (performance guarantee of optimization within the quantum neural tangent kernel (**QNTK**) approximation) in Ref [4], we have the $$E(t)=(1-\eta K)^tE(0),$$
> where the scalar $E(t)$ refers to the loss at time $t$. Intuitively, the QNTK $K$ amounts to the inner product of the average gradient vector of VQE, i.e., $K=\sum_j(\partial_E/\partial_{\theta_j})^2$.
>
> As the PINN is trained to emulate the optimization dynamics of VQE, the resulting gradient vector is expected to deviate only slightly from the ideal case. As a result, the QNTK related to PALQO, denoted by $K'$, has the relation
> $$K'=K+\delta K,$$
> where $\delta K$ depends on $\epsilon_0$ and the number of trainable parameters. Meanwhile, the training dynamics of PALQO-enabled VQE yields
> $$E'(t)=(1-\eta K')^tE(0).$$
> In this regard, we can bound the difference between estimated and ideal dynamics, i.e., $E'(t)$ and $E(t)$. That is, we have  $$\|E'(t) - E(t)\|\leq t\eta\|\delta K\|(\max(|1-\eta K|,|1-\eta K'|))^{t-1}|E(0)|.$$
> According to Theorem 3.5 in Ref [3] and our assumptions, $\delta K$ can be arbitrarily small by increasing the PINN width and depth, and $\max(|1-\eta K|,|1-\eta K'|)=1-\eta \min(K,K')<1$.
>
> Taken together, the achieved results suggest that, given a sufficiently large number of iterations $t$ and ample training data, the optimization process guided by overparameterized PALQO can converge to the global minimum of the overparameterized VQE model.
>
> [1] Di Luo, Jiayu Shen, Rumen Dangovski, and Marin Soljacic. Quack: accelerating gradient-
> based quantum optimization with koopman operator learning. Advances in Neural Information
> Processing Systems, 36, 2024.
>
> [2] Guillaume Verdon, Michael Broughton, Jarrod R McClean, Kevin J Sung, Ryan Babbush,
> Zhang Jiang, Hartmut Neven, and Masoud Mohseni. Learning to learn with quantum neural
> networks via classical neural networks. arXiv preprint arXiv:1907.05415, 2019.
>
> [3] Zeinhofer M, Masri R, Mardal K A. A unified framework for the error analysis of physics-informed neural networks[J]. IMA Journal of Numerical Analysis, 2024: drae081.
>
> [4] Junyu Liu, Francesco Tacchino, Jennifer R Glick, Liang Jiang, and Antonio Mezzacapo. Representation learning via quantum neural tangent kernels. PRX Quantum, 3(3):030323, 2022.
>
> [5] Guillaume Verdon, Michael Broughton, Jarrod R McClean, Kevin J Sung, Ryan Babbush,
> Zhang Jiang, Hartmut Neven, and Masoud Mohseni. Learning to learn with quantum neural
> networks via classical neural networks. arXiv preprint arXiv:1907.05415, 2019.

---

> ### Author Response · Authors · 2025-08-04
>
> Dear Reviewer KUyt,
>
> We hope this message finds you well. As the rebuttal deadline is approaching, we would greatly appreciate your valuable feedback. Your insights are highly important to us, and we are eager to address your comments thoroughly. We would like to kindly ask whether our latest responses have sufficiently addressed your concerns and clarified the points raised in your previous question. If there are any remaining issues or if further clarification would be helpful, we would be glad to provide additional explanations to support constructive dialogue and continued improvement.
>
> Best regards, Authors

---

> > ### Comment · Reviewer_KUyt · 2025-08-07
> >
> > I would like to thank the authors for the responses. My concerns are adequately resolved by the response. Given the authors’ detailed improvements, I have revised my score to borderline accept.

---

> > > ### Author Response · Authors · 2025-08-07
> > >
> > > Dear Reviewer KUyt,
> > >
> > > We are truly grateful for your thoughtful and positive comments. Your recognition of the contributions and quality of our work is deeply appreciated. We also thank you for highlighting the strengths of our approach, which has been instrumental in further refining our manuscript.
> > >
> > > Best, Authors

---

### Note · Authors · 2025-08-12

Dear Area Chair and Reviewers,

We sincerely thank you for the efforts to handle our submission and the constructive feedback and valuable suggestions.

We appreciate the **positive affirmation** from Reviewers KUyt, KJmn, and 2QJF regarding our resolution to the concerns and our contributions, as reducing the measurement burden is a critical step toward enhancing scalability and utility of variational quantum algorithms (VQAs) in real-world quantum applications.

Regarding Reviewer xXH5’s the remained concern on scalability, we believe some critiques may stem from misunderstandings, which we have sought to clarify in our rebuttal on 05 Aug. *Since the reviewer then has not been active during the discussion period, we take this opportunity to concisely restate the key reasons why PALQO can address larger and more complex circuits beyond the capabilities of fully classical approaches*.

(1) PALQO is, in essence, an AI-enhanced optimizer. Even for VQAs that are difficult to simulate classically (acknowledging that tensor-network and Pauli-propagator methods can handle certain structured cases), PALQO leverages gradient information obtained from quantum hardware to **efficiently predict** the trajectory of trainable parameters, thereby avoiding explicit classical simulation of the full n-qubit state. More precisely, by learning the optimization trajectory in parameter space with a PINN, PALQO confines most quantum measurements to the data-collection phase and performs subsequent updates **classically**.

(2) Our scaling experiments as shown in Figure 3, show PALQO achieves up to a 30× speedup across system sizes, highlighting the potential for large-scale quantum optimization. Besides, we have evaluated PALQO across different ansatz(e.g. HEA, HVA, and UCCSD), and the results consistently show its outperformance in measurements. We believe our work, to the best of our knowledge, is the first to address training efficiency over 30-qubit quantum systems and to empirically demonstrate the advantages in reducing measurement overhead.

(3) Corollary 3 and the added convergence analysis, acknowledged by Reviewer KUyt, mark a first step toward theory-driven design for deep learning–enhanced VQEs, meanwhile the gap between theory and practice is well-recognized in the deep learning community.

**We hope our detailed responses address the concerns raised, as they have for other reviewers. Thank you for your consideration and support.**

Best regards, Authors

---

### Decision · Program_Chairs · 2025-09-17

**Decision:**

Accept (poster)

**Comment:**

The authors present a method for accelerating VQE that dramatically reduces the number of expensive evaluations needed on quantum hardware to achieve a good estimate of the ground state of a given Hamiltonian. Due to the no-cloning theorem, analytic gradients cannot be computed on quantum hardware, and instead a method similar to finite differences is used, which becomes expensive as the number of parameters grows. By augmenting the quantum updates with the results from the PINNs, the number of parameter updates using real quantum hardware is reduced by nearly an order of magnitude. This is somewhat reminiscent of synthetic gradients for neural networks, using a learned function to replace gradient calculations. I should note that for the systems investigated (e.g. LiH, BeH2), accurate calculations can be performed with _zero_ expensive calls to quantum hardware. However, I recognize that comparison against classical computation is out of the scope of this paper, and it is assumed that at some point, quantum hardware will be able to scale beyond what can be done with classical hardware. While it remains to be seen if PALQO can scale to the point where quantum hardware has a real advantage over classical hardware, the results on the scale achievable today are promising and the reviewers mostly liked the paper. I recommend acceptance.